# Neurotrophins induce fission of mitochondria along embryonic sensory axons

Lorena Armijo-Weingart[1], Andrea Ketschek[1], Rajiv Sainath[1], Almudena Pacheco[1], George M Smith[2], Gianluca Gallo[1]*

[1]Department of Anatomy and Cell Biology, Shriner Hospitals Pediatric Research Center, Temple University Lewis Katz School of Medicine, Philadelphia, United States; [2]Department of Neuroscience, Shriner Hospitals Pediatric Research Center, Temple University Lewis Katz School of Medicine, Philadelphia, United States

**Abstract** Neurotrophins are growth factors that have a multitude of roles in the nervous system. We report that neurotrophins induce the fission of mitochondria along embryonic chick sensory axons driven by combined PI3K and Mek-Erk signaling. Following an initial burst of fission, a new steady state of neurotrophin-dependent mitochondria length is established. Mek-Erk controls the activity of the fission mediator Drp1 GTPase, while PI3K may contribute to the actin-dependent aspect of fission. Drp1-mediated fission is required for nerve growth factor (NGF)-induced collateral branching in vitro and expression of dominant negative Drp1 impairs the branching of axons in the developing spinal cord in vivo. Fission is also required for NGF-induced mitochondria-dependent intra-axonal translation of the actin regulatory protein cortactin, a previously determined component of NGF-induced branching. Collectively, these observations unveil a novel biological function of neurotrophins; the regulation of mitochondrial fission and steady state mitochondrial length and density in axons.

*For correspondence:
tue86088@temple.edu

Competing interests: The authors declare that no competing interests exist.

## Introduction

Neurotrophins are secreted growth factors that activate Trk and p75 receptors to control a multitude of aspects of neuronal cell biology (*Reichardt, 2006*). The promotion of axon collateral branching is an important effect of neurotrophin function during both development and nervous system repair (*Diamond et al., 1987*; *Diamond et al., 1992*; *Cohen-Cory, 1999*; *Schmidt and Rathjen, 2010*; *Onifer et al., 2011*; *Bilimoria and Bonni, 2013*). In the context of axon branching, neurotrophins regulate both the actin filament and microtubule cytoskeleton (*Kalil and Dent, 2014*; *Armijo-Weingart and Gallo, 2017*) and also drive the intra-axonal translation of mRNAs coding for actin regulatory proteins involved in branching (*Spillane et al., 2012*; *Batista and Hengst, 2016*). Recent work has identified mitochondria as fundamental elements in the mechanism of branching (*Courchet et al., 2013*; *Spillane et al., 2013*; *Tao et al., 2014*; *Sainath et al., 2017a*; *Wong et al., 2017*; *Smith and Gallo, 2018*). During NGF-induced branching mitochondrial respiration serves to locally drive axonal actin dynamics and intra-axonal translation of actin regulatory proteins required for NGF-induced branching (*Ketschek and Gallo, 2010*; *Spillane et al., 2012*; *Spillane et al., 2013*). Prior work has determined that neurotrophins can regulate stalling at sites of localized NGF-signaling, mitochondrial membrane potential and coordinate axonal actin dynamics and mitochondria positioning (*Chada and Hollenbeck, 2004*; *Verburg and Hollenbeck, 2008*; *Ketschek and Gallo, 2010*; *Spillane et al., 2013*; *Sainath et al., 2017a*). However, the full spectrum of the regulation of mitochondria by neurotrophins remains to be determined.

Mitochondria are dynamic organelles that undergo fission and fusion in axons (*Amiri and Hollenbeck, 2008*; *Saxton and Hollenbeck, 2012*; *Smith and Gallo, 2018*). The balance of fission and fusion serves to maintain cellular health, and fission positively contributes to aspects of cell division and organismal development. However, fission is also a characteristic response to a variety of insults and is generally considered to be a component of the cell death mechanism that when left unchecked results in fragmentation of mitochondria and loss of function (*Balog et al., 2016*). Dysregulation of the fission-fusion balance is also considered to underlie certain neuropathies and to have detrimental effects on neuronal health (*Bertholet et al., 2016*). We now report that neurotrophins induce the rapid fission of axonal mitochondria that is then followed by a persistent new steady stage of mitochondria length and density in axons. The fission of mitochondria occurs through a Drp1 GTPase and actin filament based mechanism coordinated by PI3K and MAPK signaling and it is required for the induction of axon collateral branches. This study thus unveils a novel biological function of neurotrophins, the induction of mitochondrial fission and maintenance of steady state mitochondrial length and density in axons, and provides the first example of the 'beneficial' regulation of mitochondria fission by a family of neuronal growth factors.

## Results

### Neurotrophins induce the fission of axonal mitochondria and set new steady state mitochondria density and length

For experiments involving NGF (40 ng/mL) we used embryonic day (E) seven chicken embryo sensory neurons using culturing conditions wherein only NGF responsive TrkA positive neurons extend axons (*Guan et al., 2003*; *Ketschek and Gallo, 2010*). Briefly, at E7 only TrkA+ neurons extend axons on a laminin substratum in the absence of neurotrophins (*Guan et al., 2003*; *Ketschek and Gallo, 2010*). Imaging of fluorescently labeled mitochondria during the first 10 min after NGF treatment revealed elevated rates of fission (% mitochondria undergoing fission/10 min) without detectable effects on the rate of fusion (*Figure 1A,B*) relative to the no NGF treatment control (NGF vehicle treatment). In 80% of instances of fission one or both of the emergent mitochondria underwent transport (see *Figure 1A* for an example of transport following fission; *Figure 1—figure supplement 1* shows an example of fission without subsequent transport), regardless of NGF treatment (p=0.56, Fisher's test). Thus, fission correlates with an increased probability of subsequent transport regardless of NGF treatment. Fission occurred along stalled mitochondria and we did not observe moving mitochondria undergoing fission. The proportion of mitochondria undergoing transport after fission was 110% greater than that of other mitochondria that were initially stalled but then underwent transport without having undergone fission (80% following fission relative to 38% not having undergone fission; p=0.0025. Fisher's test). Considering the whole population of axonal mitochondria, during the first 10 min of NGF treatment there was a trend toward increased transport as reflected in a slight decrease in the % of mitochondria that remained stalled throughout the imaging (*Figure 1—figure supplement 2A*), likely due to the proportion having undergone fission. The percent of time that mitochondria which underwent movement spent moving was not affected by NGF for either anterograde or retrograde directions (*Figure 1—figure supplement 2A* inset). For mitochondria having undergone fission after NGF treatment the transport was not biased in the anterograde or retrograde direction (p=0.29; Fisher's test against a hypothetical population of the same sample size exhibiting 50/50 directionality) and the velocity of movement in either direction was not affected by NGF treatment (*Figure 1—figure supplement 2B*). The velocity of transport in either direction was also not affected by NGF between moving mitochondria not observed to have undergone fission and those that underwent fission prior to transport (*Figure 1—figure supplement 2B*). Analysis of the 'run length', the net distance traveled by mitochondria during a continuous period of transport, did not reveal an effect of NGF (*Figure 1—figure supplement 2C*), although in the no NGF group we observed a trend toward more mitochondria undergoing continuous runs longer than 35 microns in either direction. However, following NGF treatment 55% of mitochondria undergoing runs exhibited one or more switches in direction during the run, compared to 15% in the no NGF group (*Figure 1—figure supplement 2D*; n = 47 and 38 respectively, p=0.0049, Fisher exact test). Thus, NGF treatment results in a bout of fission following which one or both of the two emergent mitochondria undergo redistribution within the axon, the latter correlating with fission and not linked to NGF-

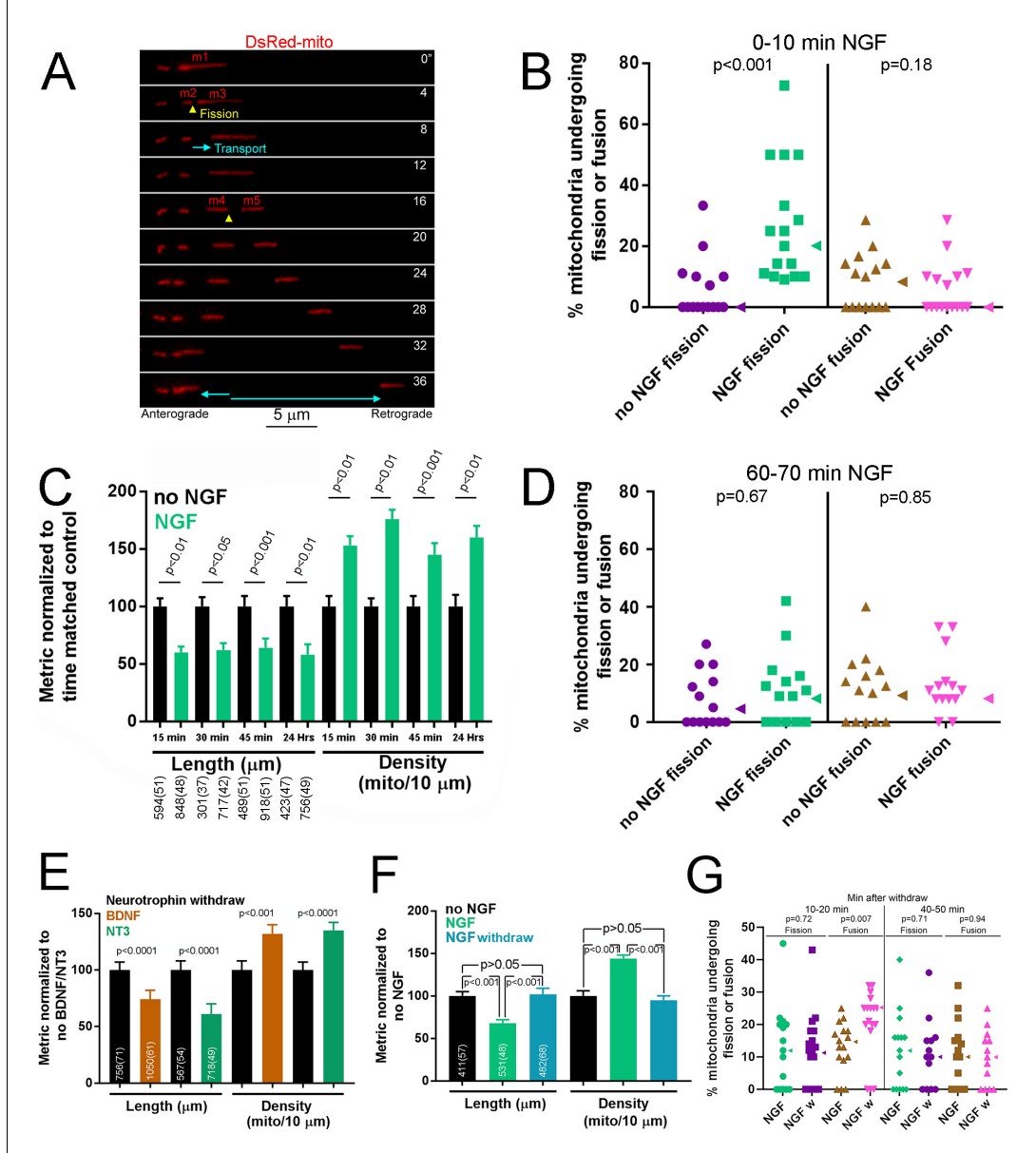

**Figure 1.** NGF induces a rapid bout of fission followed by maintenance of a new steady state of mitochondria length and density. (**A**) Example of mitochondria labeled by expression of mitochondrially targeted DsRed (DsRed-mito) undergoing fission and subsequent transport during the first 5–10 min of NGF treatment (40 ng/mL throughout). By 4' the mitochondrion labeled m1 undergoes fission giving rise to m2 and m3. The two emergent mitochondria move away from each other and at 16' m3 fissions again to give rise to m4 and m5, and again both move away from each other. (**B**) Determination of the percentage of mitochondria that underwent fission or fusion during an initial 10 min treatment with NGF. Each data point reflects one axon (n = 95 and 100 mitochondria from 15 and 16 axons for no NGF and NGF groups respectively; Mann-Whitney test). Arrowheads to the right of data points denote the median. (**C**) Mitochondria length and densities in distal 50 μm of axons, excluding growth cones, after multiple durations of NGF treatment (sample sizes shown below bars using an x(y) format where x and y denote the number of mitochondria and axons respectively). Mean and SEM shown, Bonferroni posthoc tests. (**D**) Percentage of mitochondria that underwent fission or fusion during a 10 min period after a 1 hr treatment with NGF (n = 110 and 126 mitochondria from 14 axons/group for no NGF and NGF respectively; Mann-Whitney test). Arrowheads to the right of data points denote the median. (**E**) A 45 min treatment with either BDNF or NT3 (40 ng/mL for both) decreases mitochondria length and increases density (n shown in bars using the same format as panel C; Mean and SEM shown; Welch t-test for densities, Mann-Whitney test for length). (**F**) Removal of NGF with inclusion of a function blocking NGF antibody after a 30 min treatment (NGF withdraw) restores mitochondria length and density to no NGF treatment levels (no NGF) by 3.5 hr. In contrast, in the continued presence of NGF (NGF) mitochondria exhibit the same trends as expected based on the data in panel C. (n shown in bars using the same format as panel C; Bonferroni posthoc tests for density and Dunn's posthoc tests for length). (**G**) Percentage of mitochondria that underwent fission or fusion during a 10 min period starting at 10 min or 40 min after NGF

*Figure 1 continued on next page*

*Figure 1 continued*
withdraw (NGF w) following an initial 30 min treatment with NGF. The time matched control groups (NGF) underwent similar medium exchanges as the NGF w group but the medium contained NGF. Mann-Whitney tests.
The online version of this article includes the following figure supplement(s) for figure 1:

**Figure supplement 1.** Examples of mitochondria fission, fusion and overlap without subsequent fusion.
**Figure supplement 2.** Net mitochondria transport and examples of neurotrophin treated axonal mitochondria.

treatment. Although NGF did not impact a variety of aspects of mitochondria transport, in response to NGF the mitochondria undergoing transport exhibited more frequent reversal of directionality during a continuous run.

Consistent with the bout of fission observed in live imaging studies during the first 10 min of NGF treatment, analysis of the density and length of axonal mitochondria in axons at 15–45 min following treatment with NGF revealed increased density (mitochondria/unit length of axon) of shorter mitochondria, an effect persistent after 15 min post treatment and also observed when neurons were cultured in the presence of NGF overnight (*Figure 1C* and *Figure 1—figure supplement 2E*). Consistent with a maintained new steady state of length and density by 15 min after NGF treatment, analysis of the rates of fission and fusion at 1 hr post NGF treatment (during a 10 min period for direct comparison to *Figure 1B*) did not reveal an effect of NGF (*Figure 1D*). In no NGF and 1 hr NGF conditions the rates of fission matched those of fusion, as expected for maintenance of the population of mitochondria length at steady state. Considering the data for fission at 0–10 min and 60–70 min in no NGF treatment and that in the 60–70 min in NGF treatment; a Kruskal-Wallis ANOVA with Dunn's multiple comparisons tests yields p=0.62 for the ANOVA and p>0.05 for both comparison of fission data between 0 and 10 min and 60–70 min in no NGF treatment, and for 60–70 min of NGF treatment relative to the 0–10 no NGF treatment. Thus, the initial burst of fission during the 0–10 min of NGF treatment is no longer detectable by 1 hr after NGF treatment and the percentage of mitochondria undergoing fission has returned to baseline (no NGF treatment) levels, which do not differ at 0–10 and 60–70 min.

All members of the neurotrophin family signal through Trk and p75 receptors and activate the same signaling pathways. To determine whether the effects of NGF on axonal mitochondria length and density are conserved throughout the neurotrophin family we assessed the effects of brain derived neurotrophic factor (BDNF) and neurotrophin-3 (NT3). Unlike with TrkA+ neurons there are no culturing conditions that allow us to have responsive neurons that are however naïve to the neurotrophin (*Guan et al., 2003*; *Ketschek and Gallo, 2010*). Therefore, to address the effects of BDNF and NT3 we used a protocol we established previously to generate cultures wherein only BDNF or NT-3 responsive neurons would extend axons (*Gallo et al., 1997*; *Gallo and Letourneau, 1998*). By E9 chicken sensory neurons of all classes have become neurotrophin dependent for their survival. Thus, we cultured overnight in the relevant neurotrophin, which selects the responsive neuron population, on laminin followed by a 2 hr neurotrophin deprivation period prior to restoration of BDNF/ NT3 (*Gallo et al., 1997*; *Gallo and Letourneau, 1998*). This results in selection of the relevant neuronal populations that although not naïve to the neurotrophin have a high degree of responsiveness due to neurotrophin withdraw. As with NGF, a 45 min treatment with BDNF or NT3 increased the density of axonal mitochondria and decreased their length (*Figure 1E* and *Figure 1—figure supplement 2F*).

To determine whether the effects of neurotrophins on the regulation of steady state mitochondria density and length require continuous neurotrophin treatment, we cultured E7 DRG neurons in the absence of NGF overnight and then treated with NGF for 30 min, by which point the new steady state is established (*Figure 1C*), and then either the NGF containing medium was replaced with medium containing no NGF and also including a function blocking NGF antibody or with medium containing NGF (continuous NGF treatment with sham medium exchange). Following NGF withdraw mitochondria length and density increased and decreased, respectively, back to NO NGF treatment levels by 3 hr (*Figure 1F*). Analysis of the percentage of mitochondria undergoing fission or fusion (as in *Figure 1B,D*) after NGF withdraw in the same paradigm as in *Figure 1F* revealed increased rates of fusion at 10–20 min after NGF withdraw which returned to baseline by 40–50 min after NGF withdraw (*Figure 1G*). NGF withdraw did not affect rates of fission at either time points. Thus, loss

of NGF signaling after a period of NGF signaling results in a bout of fusion. However, by 40–50 min after NGF withdraw the rates of fission and fusion are comparable (p=0.91, Mann-Whintey test), consistent with the maintenance of steady state induced by NGF withdraw. Collectively, these data reveal that NGF induces the fission of axonal mitochondria and determines a new steady state of length and density that is then maintained as long as NGF is present.

The time course of fission in response to bath application of NGF (<10 min) indicates that the response is mediated locally within the axon shaft. To specifically address whether NGF is acting locally to induce fission we locally perfused axons with NGF (*Figure 1—figure supplement 2G*). During the first 10 min of local perfusion with NGF an average of 30% of mitochondria underwent fission (*Figure 1—figure supplement 2H*), similar to bath application (*Figure 1B*). As with bath application, we did not observe an effect of local perfusion with NGF on the percentage of mitochondria undergoing fusion (*Figure 1—figure supplement 2H*). As an alternative approach to locally expose distal axons to NGF we used microfluidic compartmentalized chambers and measured the length of mitochondria in axons that grew into compartments containing no NGF or NGF (the cell body compartments did not contain NGF). Axons in compartments containing NGF exhibited shorter mitochondria relative to no NGF (*Figure 1—figure supplement 2I*), consistent with a local action of NGF.

A dose response curve showed that the effect of NGF on mitochondria length and density became evident at 10 ng/mL NGF (*Figure 1—figure supplement 2J*), consistent with activation of TrkA receptors. To further address the issue we pretreated with the TrkA inhibitor k252a prior to NGF treatment resulting in inhibition of the NGF-induced decrease in mitochondria length (*Figure 1—figure supplement 2K*). All neurotrophins bind the p75 receptor but exhibit selectivity for Trk receptors. Treatment with BDNF (100 ng/mL) did not alter mitochondria length (*Figure 1—figure supplement 2K*). Collectively, these data indicate the effects of NGF on axonal mitochondria are mediated through the TrkA receptor.

## The Drp1 GTPase mediates NGF-induced fission

Drp1 mediates fission in toxic scenarios as well as baseline fission in a variety of cell types (*Zemirli et al., 2018*). Oligomers of Drp1 accumulate at the site of fission and form a ring around the mitochondrion (*Hatch et al., 2014*). As revealed by immunocytochemistry endogenous Drp1 was found throughout axons exhibiting a punctate distribution and puncta of Drp1 colocalized with the ends of closely apposed mitochondria that may be reflective of having undergone fission around the time of fixation (*Figure 2—figure supplement 1A, B*). To directly determine if NGF-induced fission is associated with the formation of focal accumulations of Drp1, we imaged the dynamics of eYFP-Drp1 following NGF treatment. In non-fissioning mitochondria, small transient spots of eYFP-Drp1 were observed along the mitochondria consistent with prior investigations (*Ji et al., 2015*). Sites of eventual fission were demarcated by the prior focal accumulation of Drp1 (*Figure 2A*). Treatment with NGF increased the frequency of Drp1 focal accumulation along mitochondria (*Figure 2B*), consistent with its role in promoting fission. Inhibition of Drp1 activity using the pharmacological inhibitor mDivi-1 (*Figure 2C*; *Cassidy-Stone et al., 2008*; *Smith and Gallo, 2017*), the P110 cell permeable Drp1 inhibitory peptide (*Figure 2C*; *Qi et al., 2013*) or expression of dominant negative (DN) Drp1 (*Figure 2D*, inset shows an example of DNDrp1 expression and targeting to mitochondria; *Stojanovski et al., 2004*; *Barsoum et al., 2006*) blocked the effects of NGF on mitochondria fission. Overnight expression of DNDrp1 in the absence of NGF increased mitochondria lengths and decreased densities (*Figure 2E*), as expected.

Similar to the effects of removing NGF, which by 3 hr restores mitochondria to no NGF conditions (*Figure 1F*), pharmacological inhibition of Drp1 for 3 hr in cultures raised in NGF overnight also increased mitochondria length and decreased density (*Figure 2—figure supplement 1C-E*), indicating that Drp1 is required for maintenance of the NGF-induced steady state. Drp1 can also regulate the fission of peroxysomes (*Schrader et al., 2016*). However, NGF treatment did not affect the density of peroxisomes in either axons or axon branches (*Figure 2—figure supplement 1F*), indicating that the effects of NGF are specific to mitochondria.

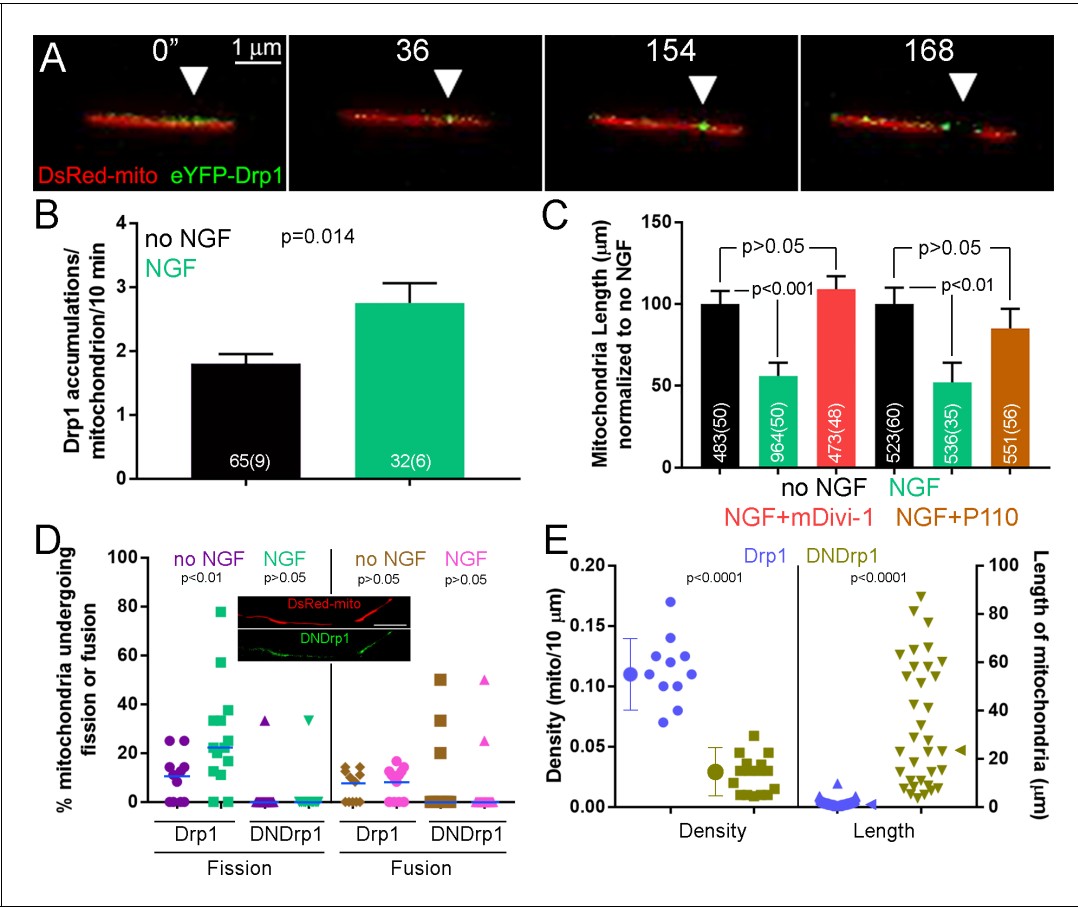

**Figure 2.** The Drp1 GTPase mediates NGF-induced fission. (**A**) Example of the formation of eYFP-Drp1 puncta (example from NGF treated axons). eYFP-Drp1 is initially heterogeneously distributed along the mitochondrion (0'). By 36 s Drp1 has begun to accumulate at the future site of fission, denoted by the arrowhead in all panels. During the ensuing period the accumulation of Drp1 becomes more focal and pronounced culminating with fission occurring at 154 s and the mitochondria moving apart by 168 s. (**B**) Quantification of the number of eYFP-Drp1 accumulations per mitochondrion during the first 10 min after treatment with NGF. n = mitochondria(axons) shown in bars, Mann-Whitney test. (**C**) Determination of mitochondria lengths with a 15 min pretreatment with 20 μM mDivi-1 followed by a 45 min treatment with NGF. Throughout this work mDivi-1 was used at 20 μM as this concentration inhibits fission without impacting complex I of the respiratory chain (*Bordt et al., 2017*; *Smith and Gallo, 2017*). Determination of mitochondria lengths after a 30 min pretreatment with 5 μM P110 followed by a 30 min treatment with NGF. For all pharmacological experiments in this report treatment with vehicle (DMSO) was performed as control. n = mitochondria(axons) shown in bars; Dunn's posthoc tests. (**D**) Live imaging analysis of the rates of fission (% mitochondria/10 min) before and after NGF treatment (0–10 min) in the axons of eYFP-Drp1 (Drp1) or eYFP-DNDrp1 (dominant negative Drp1; DNDrp1) expressing neurons. Mitochondria were labeled through co-expression of mitochondrially targeted DsRed. Each data point reflects one axon. Mann-Whitney tests. Blue lines denote median. The inset shows and example of DNDrp1 expression. As with wild type Drp1 DNDrp1 targeted to mitochondria. Bar = 5 μm. (**E**) Mitochondria length and densities in the axons of neurons expressing Drp1 or DNDrp1 in the no NGF condition prior to NGF treatment. Data points for density represent axons and for lengths individual mitochondria within those axons. Mann-Whitney test and Welch t-test for length and density comparisons respectively. Mean and SD are shown to the left of data points for density, and median is denoted by arrowheads to the right of data points for length.

The online version of this article includes the following figure supplement(s) for figure 2:

**Figure supplement 1.** Involvement of Drp1 in mitochondria fission but not the regulation of peroxysomes.

## Patches of actin filaments at sites of fission mediate Drp1-dependent NGF-induced fission

Actin filaments are emerging as a component of the fission mechanism (*Hatch et al., 2014*; *Kraus and Ryan, 2017*), and NGF upregulates axonal actin dynamics reflected in localized dynamic patches of filaments (*Loudon et al., 2006*; *Ketschek and Gallo, 2010*; *Spillane et al., 2011*; *Spillane et al., 2012*). Imaging of eYFP-β-actin revealed that sites of fission correlated with the presence of actin patches along the mitochondrion (*Figure 3A*). We observed 80% of fission sites

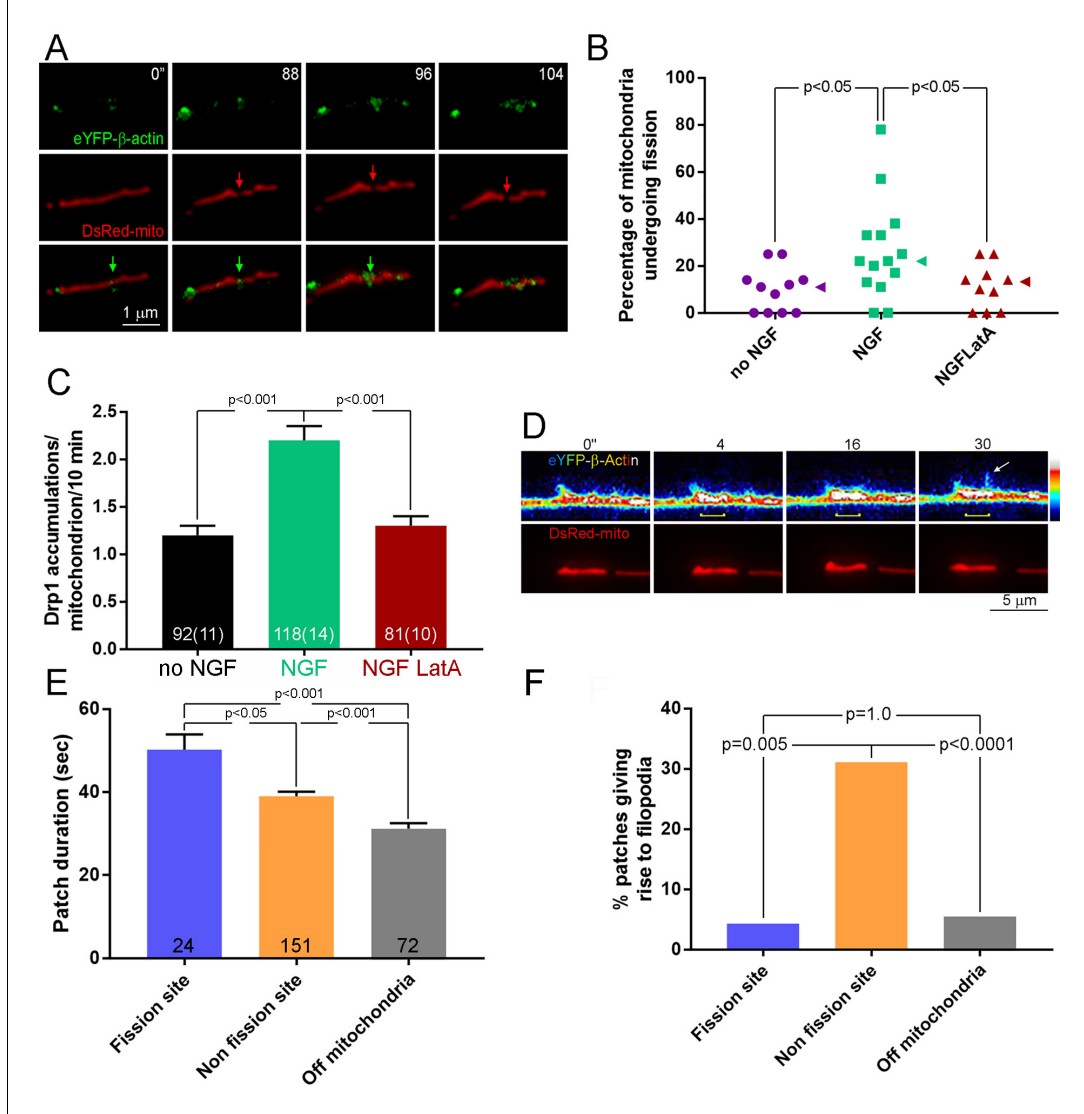

**Figure 3.** Patches of actin filaments mediate NGF-induced fission. (**A**) Timelapse sequence showing the formation of an actin filament patch at the site of fission during the first 10 min of NGF treatment. The initiation of the actin patch is shown at time 0 s (green arrow) and tracked throughout the process of fission. The patch elaborates at the future site of fission (red arrow) that becomes evident at 96 s. Other actin patches formed along the mitochondrion are not associated with sites of fission. (**B**) Pretreatment (15 min) with the actin depolymerizing drug Latrunculin A (LatA; 4 μM) prior to NGF treatment (0–10 min) blocks the NGF-induced increase in the rate of mitochondria undergoing fission (% mitochondria/10 min). Each data point reflects one axon. Dunn's posthoc multiple comparison tests. NGF and no NGF treatments included treatment with the DMSO vehicle. Median is denoted by arrowheads to the right of data points. (**C**) LatA pretreatment blocks the NGF-induced increase in the formation of Drp1 accumulations along mitochondria. Same experimental design as in (**B**), n = mitochondria(axons) shown in bars. Dunn's posthoc multiple comparison tests. (**D**) Example of actin patch formation associated with a mitochondrion that gives rise to the emergence of a filopodium but not fission. By 4 s an actin patch has begun forming at the site populated by the mitochondrion (yellow bracket at 4–30 s). At 30 s a filopodium emerges from the patch (arrow). The heat map ranges 0–4095 in pixel intensities. (**E**) Analysis of the duration of actin patches during the first 10 min of NGF treatment as a function of their relationship to mitochondria and sites of fission. n = number of patches from 19 axons. Dunn's posthoc multiple comparison tests. (**F**) Analysis of the probability that an actin patch will give rise to a filopodium as a function of their relationship to mitochondria and sites of fission. Same data set as in (**E**). Fisher's exact tests.

The online version of this article includes the following figure supplement(s) for figure 3:

**Figure supplement 1.** Requirement of Arp2/3 nucleated actin filaments in NGF-induced fission.

(n = 25; 20/25 sites) colocalizing with detectable actin patches. The mean length of mitochondria that underwent fission was 8.3 ± 0.8 µm (mean + SEM throughout this section; *Table 1* column A). The mean summed length of all actin patches present along each mitochondrion at the first frame wherein fission was evident was 0.83 ± 0.16 µm/mitochondrion (*Table 1* column B). We next calculated the probability that the site of fission would randomly overlap with any actin patch present at the time of fission for each mitochondrion (summed length of actin patches along the mitochondrion/length of the mitochondrion; *Table 1* column C). To estimate the probability of having observed random overlap between actin patches and sites of fission in all 20 instances, we next calculated C1xC2x...C19xC20. Based on these empirical data, the probability of observing the detected the observed 20 instances of overlap between actin filament patches and sites of fission in this data set is $1.75 \times 10^{-21}$.

To address whether actin filaments colocalizing with sites of fission may have a role in fission, we depolymerized actin filaments using latrunuclin-A prior to treatment with NGF. Latrunculin-A blocked the promotion of fission by NGF (*Figure 3B* and *Figure 3—figure supplement 1A*) without affecting fusion (*Figure 3—figure supplement 1B*). Furthermore, treatment with latrunculin A blocked the NGF-induced increase in the formation of Drp1-puncta along mitochondria (*Figure 3C*). Collectively, the data indicate that the mechanism of NGF-induced fission requires the combined activity of the Drp1 GTPase and actin filaments, the latter being required for the formation of focal accumulations of Drp1 along mitochondria.

## Actin patches associated with sites of fission are distinct from those giving rise to filopodia

Axonal actin patches have a previously described role in serving as cytoskeletal platforms for the emergence of axonal filopodia (*Figure 3D*; *Loudon et al., 2006*; *Ketschek and Gallo, 2010*;

**Table 1.** Data set used for estimation of the probability of random overlap between actin patches and sites of fission along mitochondria.

| Mitochondria length (µm) | Summed length of actin patches present on mitochondrion at the time of fission (µm) | Proportion of mitochondria length occupied by actin patches (B/A) |
|---|---|---|
| 7.17 | 0.47 | 0.07 |
| 6.90 | 0.69 | 0.10 |
| 12.00 | 1.1 | 0.09 |
| 8.15 | 0.69 | 0.08 |
| 16.78 | 3.69 | 0.22 |
| 7.35 | 0.53 | 0.07 |
| 3.94 | 0.65 | 0.16 |
| 10.24 | 0.75 | 0.07 |
| 9.65 | 0.75 | 0.08 |
| 4.10 | 0.69 | 0.17 |
| 7.48 | 0.75 | 0.10 |
| 6.65 | 0.69 | 0.10 |
| 16.78 | 0.23 | 0.01 |
| 5.87 | 0.69 | 0.12 |
| 4.70 | 0.50 | 0.11 |
| 8.30 | 0.75 | 0.09 |
| 6.90 | 0.80 | 0.12 |
| 7.70 | 0.78 | 0.10 |
| 11.30 | 0.86 | 0.08 |
| 4.40 | 0.56 | 0.13 |

*Hand et al., 2015*; *Ketschek et al., 2016*) and the emergence of filopodia occurs preferentially from patches associated with mitochondria (*Figure 3D*; *Spillane et al., 2013*; *Sainath et al., 2017a*). Analysis of the duration of patches during the first 10 min of NGF treatment revealed that patches associated with sites of fission are longer in duration than those that arise at sites colocalizing with mitochondria but not associated with fission sites, and also those that arise at non-mitochondrial sites along the axon (*Figure 3E*). Consistent with a prior report (*Sainath et al., 2017a*), actin patches associated with mitochondria exhibited longer durations than those not associated with mitochondria regardless of whether they were associated with sites of fission (*Figure 3E*). Actin patches associated with mitochondria but not involved in fission exhibited an over 5-fold greater probability than patches associated with fission to give rise to a filopodium (*Figure 3F*). Patches associated with sites of fission generated filopodia with the same probability as patches not associated with mitochondria (*Figure 3F*). These data indicate that fission-associated actin patches represent a subpopulation of patches that colocalize with mitochondria and are specialized for fission and not the formation of filopodia.

Prior work addressing the cytoskeletal mechanism of actin patch formation underlying filopodia formation determined a role for the actin filament nucleating complex Arp2/3 (*Spillane et al., 2011*; *Spillane et al., 2012*). However, since the above analysis revealed differences between actin patch subpopulations we tested the role of the Arp2/3 complex in mediating NGF-induced fission. Pretreatment with an Arp2/3 inhibitor blocked the effects of NGF on mitochondria fission (*Figure 3—figure supplement 1C*), indicating that although patches associated with fission sites differ from those involved in filopodia formation they share a conserved role for Arp2/3.

## Coordination of NGF-induced fission by combined PI3K and Mek-Erk signaling

The PI3K and Mek-Erk pathways are major effectors of neurotrophin signaling and regulate neuronal morphogenesis (*Huang and Reichardt, 2003*; *Reichardt, 2006*). Along axons, PI3K activity is required for mediating the effects of NGF on axonal actin patch formation (*Ketschek and Gallo, 2010*). The activity of the Drp1 GTPase is under regulation by posttranslational modifications, and phosphorylation of S616 in Drp1 enhances its activity and fission (*Hall et al., 2014*). Additionally, previous work showed that Erk phosphorylates Drp1 at S616 in cancer cells thereby promoting fission (*Kashatus et al., 2015*; *Serasinghe et al., 2015*). Inhibition of either PI3K or Mek-Erk signaling prevented the NGF-induced increase in mitochondria fission (*Figure 4A*) and also impaired the increase in Drp1 accumulations along mitochondria induced by NGF (*Figure 4B*), but did not impact fusion rates (*Figure 4—figure supplement 1A*).

NGF induced increases in the levels of pS16-Drp1 along axons during the time of prominent fission (10 min post treatment) (*Figure 4C,D*) without altering the axonal levels of total Drp1 protein (*Figure 4—figure supplement 1B*). The increase in total intensity of pDrp1 along axons was at least in part due to a higher density of puncta of staining. Inhibition of PI3K had no effect on NGF-induced pS616 phosphorylation (*Figure 4D*). In contrast, inhibition of Mek-Erk signaling abolished the effects of NGF on pS616 levels (*Figure 4D*) without affecting the levels of total Drp1 protein (*Figure 4—figure supplement 1B*). The Mek-Erk-dependent NGF-induced increases in pS616 levels also correlated with the activation of Mek-Erk signaling in axons after NGF treatment (*Figure 4—figure supplement 1C,D*). At 7.5 min after NGF the median levels of pErk in axons were elevated by 320%, and after 30 min and 1 hr of treatment levels were elevated by 134% and 67%, respectively. The activity of PI3K in axons, as reflected in the phosphorylation and activation of Akt (at T308), occurred within 2 min of NGF treatment and was most pronounced during the first 4–8 min of NGF treatment (maximal increase of approximately 450% above control levels at 6 min; *Figure 4—figure supplement 1D,E*). By 10 min of treatment levels of pAkt declined to approximately 100% greater than control levels consistent with our prior determination that at 1 hr post NGF treatment levels of pAkt in axons are approximately increased by 100% (*Silver et al., 2014*). To address whether the PI3K-Akt and Erk pathways could impact mitochondria locally, we measured the levels of activated Akt and Erk in axon segments containing mitochondria with and without NGF treatment. NGF elevated the activity of both pathways in axon segments containing mitochondria at 6 min of treatment when fission is occurring (*Figure 4—figure supplement 1G-J*). These data indicate that the Mek-Erk pathway regulates Drp1 activity through S616 phosphorylation independent of PI3K, both pathways contribute to the recruitment of Drp1 to focal accumulations along mitochondria, and the activation

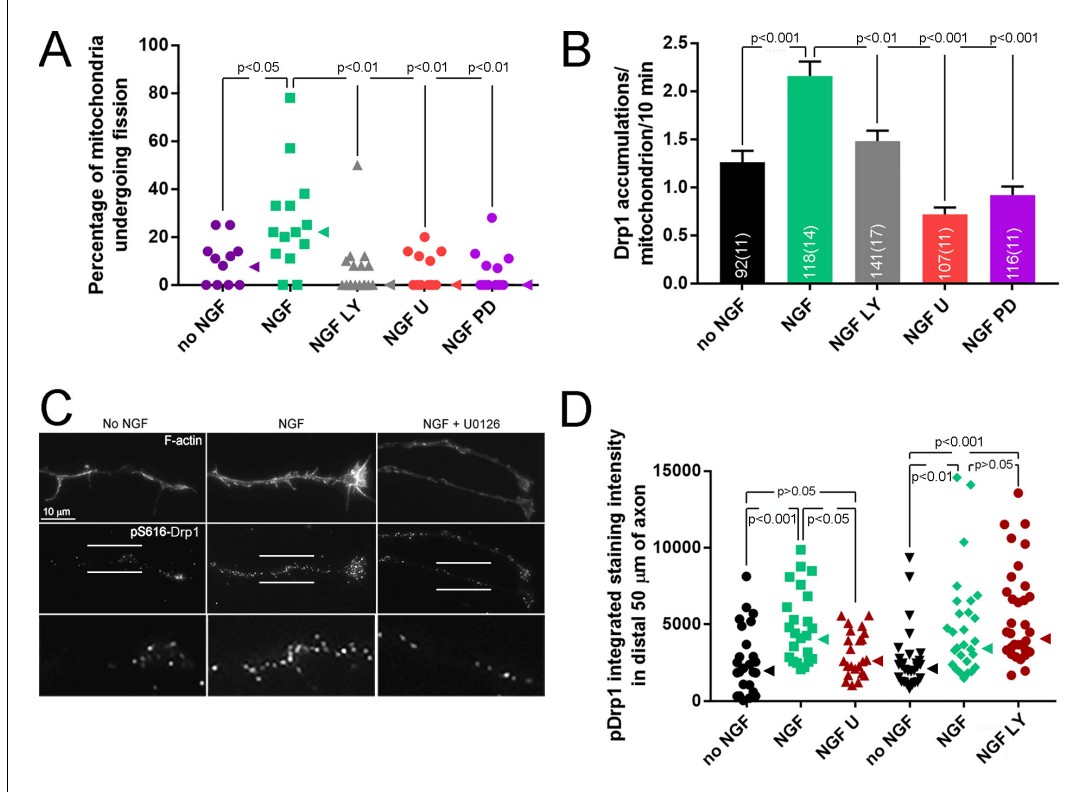

**Figure 4.** PI3K and Mek-Erk signaling are required for NGF-induced mitochondria fission. (**A**) Inhibition of PI3K (LY294002, LY, 25 μM) or Mek-Erk (U0216, U, 50 μM; PD325901, PD, 1 μM) both prevented the NGF-induced increase in the rate of mitochondria undergoing fission during the first 10 min of NGF treatment (% mitochondria/10 min). Each data point reflects an axon. The no NGF and NGF groups received the DMSO vehicle and are the same as shown in *Figure 3B* as pharmacological experiments were performed in parallel. Dunn's posthoc multiple comparison tests performed using no NGF, NGF and NGF+drug group within drug treatment. Median is denoted by arrowheads to the right of data points. (**B**) LY and U/PD pretreatment blocks the NGF induced increase in the formation of Drp1 accumulations along mitochondria. Same experimental design as in (**A**), n = mitochondria(axons) shown in bars. Dunn's posthoc multiple comparison tests performed using no NGF, NGF and NGF+drug group within drug treatment. (**C**) Examples of axons stained with anti-pS616-Drp1 antibodies and phalloidin to reveal actin filaments (F-actin). The pS616-Drp1 staining pattern was punctate. NGF elevated the staining levels and pretreatment with U prevented the NGF-induced increase. The bottom panels show empty magnification examples of the axonal domains denoted by the parallel lines in the pS616-Drp1 panels above. For presentation purposes, all images in panel were equally digitally brightened to enhance visual appreciation of the signal. (**D**) Quantification of the total intensity of pS616-Drp1 staining in distal axons. Each datum reflects one axon, Dunn's posthoc multiple comparison tests within drug treatment experiment. Median is denoted by arrowheads to the right of data points.

The online version of this article includes the following figure supplement(s) for figure 4:

**Figure supplement 1.** Signaling mechanism of NGF-induced fission.

of the pathways is highest during the period of increased rates of fission and occurs within axon segments containing mitochondria.

To determine if either the PI3K-Akt or Mek-Erk pathways contribute to the maintenance of the NGF steady state, we treated with NGF for 30 min followed by a 2 hr treatment with inhibitors of PI3K or Mek-Erk, as in prior Figures, in the continued presence of NGF. Inhibition of either pathway increased mitochondria length (*Figure 4—figure supplement 1K*) indicating both are required for the maintenance of the NGF-induced steady state.

## Drp1-mediated fission is required for NGF-induced axon branching

Mitochondria have emerged as important mediators of axon branching (*Courchet et al., 2013*; *Tao et al., 2014*; *Sainath et al., 2017a*; *Wong et al., 2017*; *Smith and Gallo, 2018*), including NGF-induced branching (*Spillane et al., 2013*). Branching in response to NGF treatment is statistically detectable at 15 min and maximal at 30 min (*Spillane et al., 2012*). The initial burst of fission in response to NGF occurs by <15 min post-treatment and by 15 min the new steady state of

mitochondria density and length is established. PI3K is involved in multiple aspects of the branching mechanism activated by NGF (*Ketschek and Gallo, 2010*; *Spillane et al., 2012*). Similarly, inhibition of Mek-Erk signaling also blocks NGF-induced branching (*Figure 5—figure supplement 1A*). Considering that the fission temporally precedes the branching, we determined whether the fission is required for branching. Inhibition of Drp1 with mDivi-1, P110, or expression of DNDrp1 blocked NGF-induced axon branching (*Figure 5A–D*). To address whether the requirement for fission in NGF-induced axon branching involves axonal mitochondria, and not mitochondria that may undergo rapid transport from the cell body, we severed axons from the cell body immediately prior to treatment with NGF as performed in our prior studies addressing the role of intra-axonal protein synthesis in NGF-induced axon branching (*Spillane et al., 2012*). As previously determined, severing axons from the cell bodies does not alter branching in response to NGF (*Figure 5—figure supplement 1B*) and treatment with mDivi-1 immediately after severing and prior to NGF application blocked NGF-induced axon branching (*Figure 5—figure supplement 1B*). In contrast to the requirement of fission for NGF-induced axon branching, treatment with mDivi-1 did not alter the increases in growth cone size and number of growth cone filopodia induced by NGF (*Figure 5—figure supplement 1C, D*), indicating a role for fission specific to axon branching and not the regulation of growth cone morphology as previously also observed for NGF-induced intra-axonal protein synthesis (*Roche et al., 2009*; *Spillane et al., 2012*).

The formation of axonal filopodia is the first step in the establishment of branches (*Kalil and Dent, 2014*; *Armijo-Weingart and Gallo, 2017*). Actin patches formed along the axon serve as precursors to filopodia formation and NGF increases the rate of patch formation in a manner dependent on NGF-induced PI3K activity and intra-axonal protein synthesis (*Spillane et al., 2012*). NGF increases the rate of filopodia formation but does not affect the probability that a patch will give rise to a filopodium, and the increase in filopodia formation is due to the increase in patch formation (*Ketschek and Gallo, 2010*). Similar to the time course of NGF-induced branching, increases in net actin patch formation rates are first detectable at 15–21 min post treatment and are further increased by 30–36 min (*Spillane et al., 2012*). Analysis of the effects of a 30 min treatment with NGF on actin patches revealed that inhibition of fission using mDivi-1 blocked the NGF-induced increase in actin patch formation (*Figure 5E*). In contrast, inhibition of fission did not alter patch duration or the probability of filopodial emergence from patches (*Figure 5—figure supplement 1E, F*), two aspects of patch dynamics that are not under regulation by NGF (*Ketschek and Gallo, 2010*). Inhibition of Mek-Erk blocked the NGF-induced increase in the rate of actin patch formation (*Figure 5—figure supplement 1G*), an observation consistent with the inhibition and requirement for fission detailed above but not excluding the involvement of Erk in other mechanisms. Inhibition of Erk did not affect patch duration before or after NGF treatment (Kruskal-Wallis ANOVA, p=0.16, n = 59–142 patches/group), indicating is it not regulating net actin dynamics in patches.

To determine if NGF-induced fission may also contribute to branching through the shortening of mitochondria, we considered the lengths of mitochondria in branches relative to the main axon under conditions of NGF and NO NGF treatment. In NGF treatment conditions, the lengths of mitochondria in branches were not different that those along the main axons shaft (*Figure 5F*). In contrast, in the absence of NGF treatment mitochondria in branches that formed without NGF stimulation were shorter than in the axon shaft (*Figure 5F*). Furthermore, the lengths of mitochondria targeted to branches formed in the absence of NGF were of similar lengths to those found in axons, and branches, after NGF treatment. NGF-induced branches also exhibited increased densities of mitochondria relative to branches formed in the absence of NGF (*Figure 5G*). This observation indicates that NGF-induced fission generates a population of mitochondria that have facilitated access to nascent branches.

## Inhibition of Drp1 function impairs the developmental collateral branching of sensory axons in the spinal cord

Both exogenous and endogenous NGF can induce branching of sensory axons in vivo in the adult spinal cord (*Brown and Weaver, 2012*; *Keefe et al., 2017*). However, NGF does not control the formation of sensory axon collateral branches in the developing spinal cord (*Patel et al., 2000*). To address whether the regulation of fission is required for the non-NGF dependent developmental branching of sensory axons we used electroporation to express DNDrp1in sensory neurons in ovo (*Spillane et al., 2011*; *Spillane et al., 2012*; *Spillane et al., 2013*; *Donnelly et al., 2013*). To image

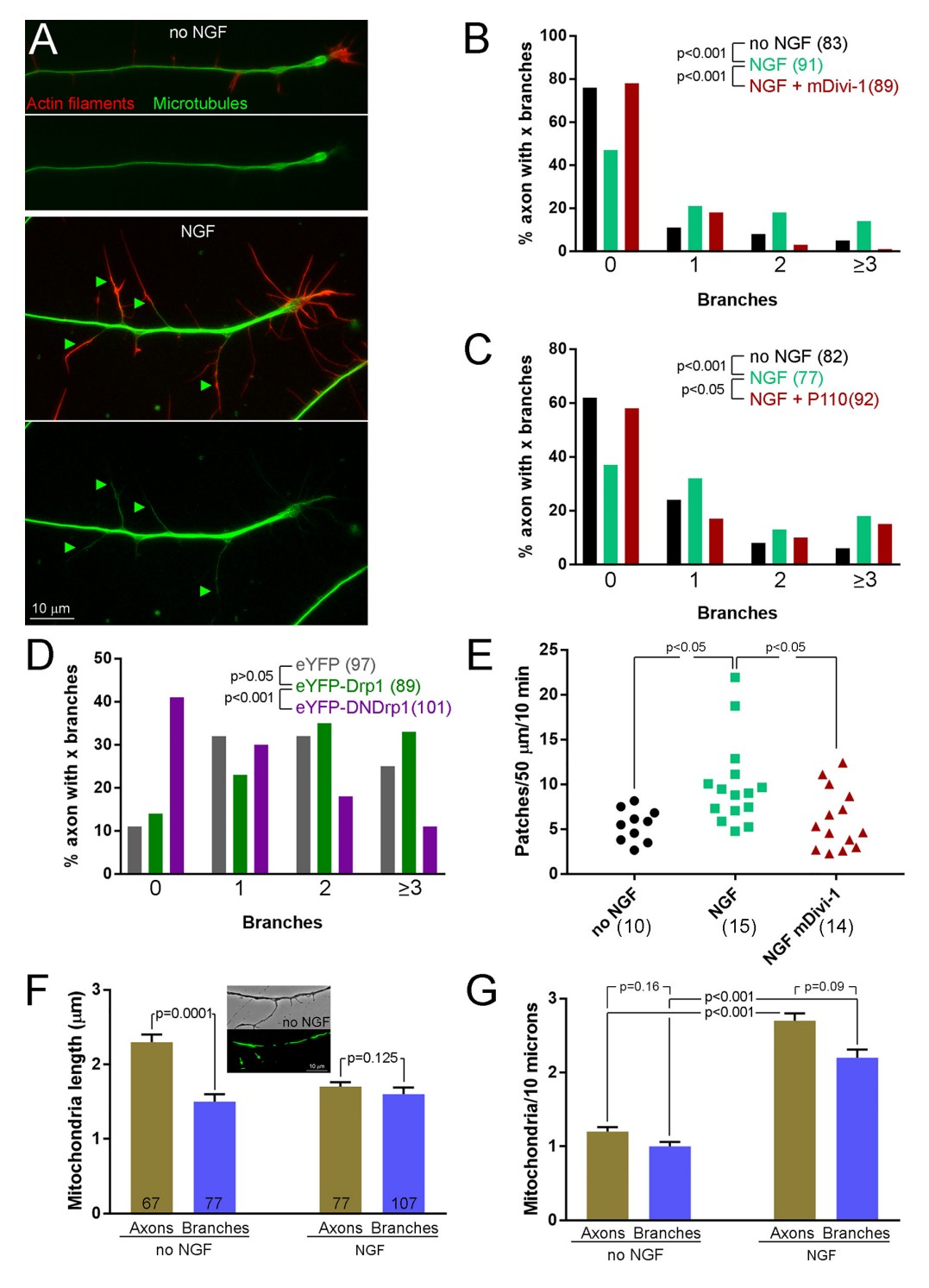

**Figure 5.** Inhibition of fission impairs NGF-induced axon branching. (**A**) Example of NGF-induced axon branching. Top panels show axonal morphology in the no NGF treatment group. Some axonal filopodia are present but not branches. Bottom panels show an NGF-treated axons with four branches (green arrowheads). Mature branches contain microtubules and distally actin filaments. (**B**) Quantification of axon branches ± mDivi-1 or DMSO pretreatment. For panels (**B**)-(**D**) n = axon numbers shown in (). Mann-Whitney test. (**C**) Quantification of axon branches ± P110 or vehicle pretreatment. Mann-Whitney test. (**D**) Quantification of axon branches in dissociated neurons cultured overnight in NGF expressing eYFP (baseline control), eYFP-Drp1 or eYFP-DNDrp1. The differences in baseline branch number reflect the use of dissociated neurons relative to explant cultures as in panels (**B**) and (**C**) for the acute NGF treatment experiments. Mann-Whitney test. (**E**) Quantification of the rates of actin patch formation ±mDivi-1 or DMSO treatment. Each data point reflects one axon and the number of axons is shown in () below the data. Dunn's posthoc multiple comparison tests.
*Figure 5 continued on next page*

*Figure 5 continued*

(F) and (G) Quantification of mitochondria length and density, respectively, in the axons and branches of axons emanating from explants raised in either no NGF or NGF overnight. n = axons shown in the bars labeled Axons; n for branches from this set of axons is denoted in the bars labeled Branches. Mean and SEM. Mann-Whitney tests.

The online version of this article includes the following figure supplement(s) for figure 5:

**Figure supplement 1.** Role of fission in branching.

axons and their branches, living spinal cords were acutely explanted and imaged ex vivo using established methods (*Spillane et al., 2011*; *Spillane et al., 2012*). Expression of DNDrp1 increased the lengths of mitochondria in axons extending in the spinal cord (*Figure 6A,B*). The mean total 'mass' of mitochondria (mean length X mean density/10 µm) was 2.67 and 5.47 for control and DNDrp1 groups (using the data presented in *Figure 6B*). An increase of total mass of mitochondria in axons when fission is impaired in vivo is consistent with a recent report focusing on cortical neurons in vivo (*Lewis et al., 2018*). The effects of expression of DNDrp1 on branching are thus not attributable to a decrease in mitochondria mass in axons. Analysis of branching along the axons of neurons

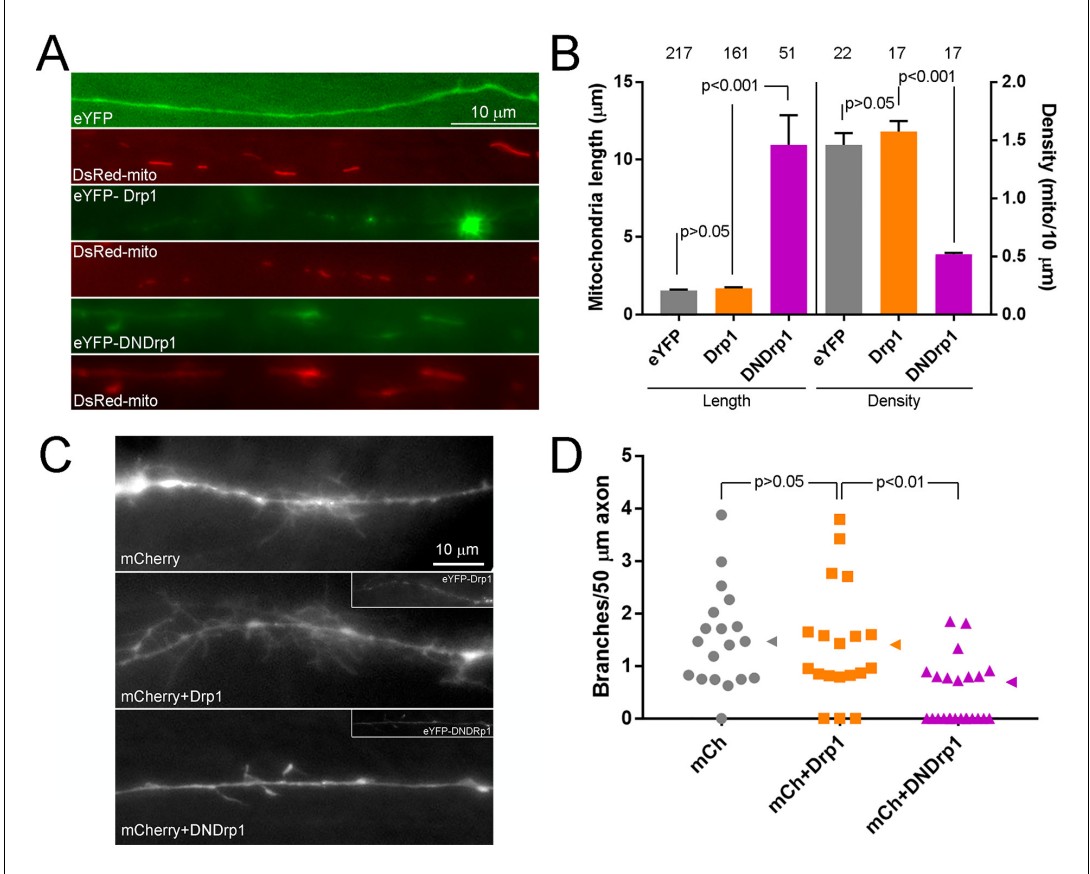

**Figure 6.** In vivo/ovo expression of DNDrp1 impair the developmental branching of sensory axons. (A) Examples of DsRed-mito labeled mitochondria in eYFP, eYFP-Drp1 or eYFP-DNDrp1 co-expressing axon in the acutely explanted living spinal cord of an E7 embryo. (B) Quantification of the length and density of mitochondria in axons in the living spinal cord as shown in panel (A). Number of mitochondria and axons are shown above the bars for length and density measurements respectively. 3–4 spinal cords/group. The sample sizes shown over the bars for density reflect the number of axons sampled. Dunn's posthoc multiple comparison tests. (C) Examples of the morphology in sensory axons extending into the E7 embryonic spinal cord. The images are maximum intensity projections from Z-stacks. Control axons expressing only mCherry, to visualize morphology, were similar to those expressing mCherry and wild type Drp1. In contrast, the axons expressing mCherry and DNDrp1 exhibited simplified morphologies. For the groups expressing Drp1 constructs the insets show the signal from the eYFP label. (D) Quantification of the number of branches (axonal projections longer than 10 µm) per unit length of axon in the living spinal cord. Each data point reflects one axon sampled from 3 to 4 spinal cords/group. Dunn's multiple comparison tests. Median is denoted by arrowheads to the right of data points.

expressing DNDrp1 revealed decreased density of branches (*Figure 6C,D*), indicating the regulation of mitochondria length and density by fission is also required for non-neurotrophin dependent branching during development. To address whether there may be endogenous changes in mitochondria length during the period of developmental sensory axon branching in the cord (E5-E10; *Eide and Glover, 1995*) we compared lengths in the axons of E7 and E14 cultured neurons (24 hr in NGF). The median length of E14 mitochondria was increased by 33% relative to that of E7 (1.6 and 1.2 μm, respectively; n = 193 and 285, p=0.0001 Mann-Whitney test).

## Inhibition of Drp1-mediated fission impairs the NGF-induced intra-axonal translation of cortactin

The induction of axon branches and increase in the formation of axonal actin patches by NGF are dependent on NGF-induced intra-axonal protein synthesis of Arp2/3 regulators and Arp2/3 subunits (*Spillane et al., 2012*; *Spillane et al., 2013*). Given that the NGF-induced fission is required for axon branching we sought to determine whether it may also be involved in mitochondria dependent NGF-induced intra-axonal synthesis of the Arp2/3 regulator cortactin (*Spillane et al., 2012*; *Spillane et al., 2013*; *Sainath et al., 2017a*). To monitor the translation of cortactin in axons, we used a previously detailed reporter (myrGFP-cortactin 3'UTR; *Spillane et al., 2012*; *Spillane et al., 2013*; *Sainath et al., 2017a*). Briefly, the 3'UTR of mRNAs is a determinant of the association of axonally targeted mRNAs with ribonuceloprotein pacrticles that transport the mRNAs into axons and regulate their translation (*Gomes et al., 2014*). Cotranslationally myristoylated GFP (myrGFP) is targeted to the nearest membrane wherein its diffusion is largely limited. Thus, constructs containing the 3'UTR of the mRNA of interest designed to express myrGFP are used in fluorescence recovery after photobleaching (FRAP) studies to monitor the local translation of specific mRNAs (*Aakalu et al., 2001*). Live imaging of the FRAP of myrGFP-cortactin 3'UTR in axons after photobleaching showed that inhibition of fission using mDivi-1 prior to NGF treatment attenuated the translation of myrGFP-cortactin 3'UTR (*Figure 7A,B*). As demonstrated by quantitative immunocytochemistry, NGF treatment also increases the protein levels of cortactin protein in axons in a manner dependent on intra-axonal protein synthesis (*Spillane et al., 2013*; *Ketschek et al., 2016*). Inhibition of fission blocked the NGF-induced increase in the axonal levels of cortactin (*Figure 7C,D*). We have previously reported the requirement for PI3K signaling in the NGF-induced intra-axonal translation dependent increase in axonal levels of cortactin (*Spillane et al., 2012*). Inhibition of Mek-Erk signaling attenuated the NGF-induced increase in cortactin axonal protein levels (*Figure 7—figure supplement 1A*). These data indicate that the fission of mitochondria in response to NGF is required to mount the intra-axonal protein synthesis of cortactin that is in turn required for NGF-induced branching.

We addressed whether protein synthesis is required for NGF-induced mitochondria fission. Pretreatment with the translational inhibitor cycloheximide, which blocks NGF-induced axonal protein synthesis and branching (*Spillane et al., 2012*), did not impact the change in mitochondria length induced by NGF (*Figure 7—figure supplement 1B*). Collectively, the data indicate that while NGF-induced mitochondria fission contributes to establishing NGF-induced intra-axonal translation the latter is not required for the former.

## Discussion

This major finding of this investigation is the determination of a novel biological action of neurotrophins; the activation of an initial burst of mitochondria fission resulting in the establishment, and subsequent maintenance, of new a steady state of mitochondria density and length of in sensory axons. Two of the major signal transduction pathways activated by neurotrophins that control morphogenesis, PI3K and Mek-Erk, coordinate to regulate the mechanism of mitochondrial fission and their activity is required to maintain the new steady state. Secondarily, the NGF-induced fission of axonal mitochondria is required for NGF-induced axon branching, revealing the first contribution of the NGF-induced fission to a cell biological process regulated by NGF. The results of the current study are placed in context with the additional established components of the mechanism of NGF-induced axon collateral branching in *Figure 8A*. The observation that neurotrophins induce mitochondria fission and establish a new steady state of length and density along axons has implications for understanding how neurotrophins impact the biology of the nervous system, and also indicate that the

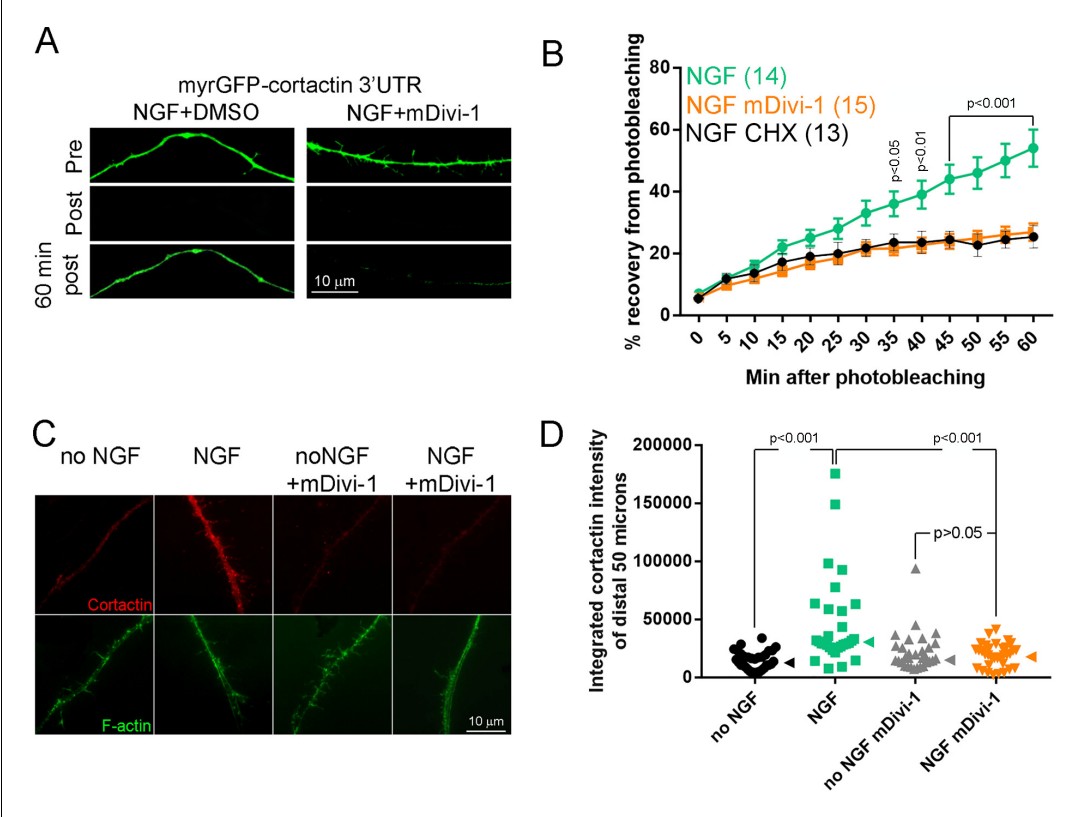

**Figure 7.** Inhibition of fission impairs the NGF-induced translation of axonally targeted cortactin mRNA. (**A**) Example of FRAP experiments showing pre-photobleaching, just post-photobleaching and 60 min post-photobleaching images. The NGF treated axon recovers fluorescence more extensively than the mDivi-1 pretreated axon. (**B**) Graph showing the quantification of the relative FRAP (% recovery from time 0 value to pre-photobleaching value) in NGF+DMSO and NGF+mDivi-1 treatment groups (n = axons). Differences between NGF+DMSO and NGF+mDivi-1 groups are detected starting at 35 min using Bonferroni posthoc time-matched multiple comparison tests. The NGF+CHX (35 µM cycloheximide 30 min pretreatment before NGF treatment as in *Spillane et al., 2012*) shows the extent of recovery due to translation independent diffusion or transport of the myrGFP reporter from the axon proximal to the region of bleaching. The recovery in the NGF+CHX and NGF+mDivi-1 groups was not different at any time point using Bonferroni posthoc time-matched multiple comparison tests. (**C**) Examples of the staining levels of cortactin in axons. A 30 min treatment with NGF increases cortactin levels and the increase is impaired by pretreatment with mDivi-1. Pretreatment with CHX blocks the increase in axonal cortactin levels induced by NGF (*Spillane et al., 2012*). (**D**) Quantification of the levels of cortactin in distal axons (total integrated staining intensity). Each datum represents one axon. Dunn's posthoc multiple comparison tests. Median is denoted by arrowheads to the right of data points.
The online version of this article includes the following figure supplement(s) for figure 7:

**Figure supplement 1.** Effects of Mek-Erk inhibition on cortactin translation and inhibition of translation of NGF-induced fission.

regulation of mitochondria length and density may be a general determinant of the competency of axons to undergo collateral branching. Consistent with this suggestion, inhibition of fission machinery in vivo suppresses the developmental branching of both sensory axons (current study) and cortical axons (*Lewis et al., 2018*), neither of which is considered to be under regulation by neurotrophins.

The field of mitochondria fission has relied on the analysis of spontaneously occurring fission or fission induced by impairing mitochondrial function (e.g., using mitochondrial poisons). A role for growth factor signaling in fission has not been reported. The mechanism of fission used by neurotrophins is Drp1 dependent, as with other forms of fission (*Kraus and Ryan, 2017*). This study reveals a role for actin filaments in fission, consistent with an emerging literature (*Hatch et al., 2014*). Moreover, in the case of neurotrophin signaling, we were able to observe the formation of discrete actin patches at sites of fission. This observation unveils a novel role for axonal actin patches, their involvement in fission, independent of their previously described role in the formation of axonal filopodia that represents the first step in the sequence of morphological changes leading to axon collateral branching (reviewed in *Armijo-Weingart and Gallo, 2017*). Furthermore, the data show that even

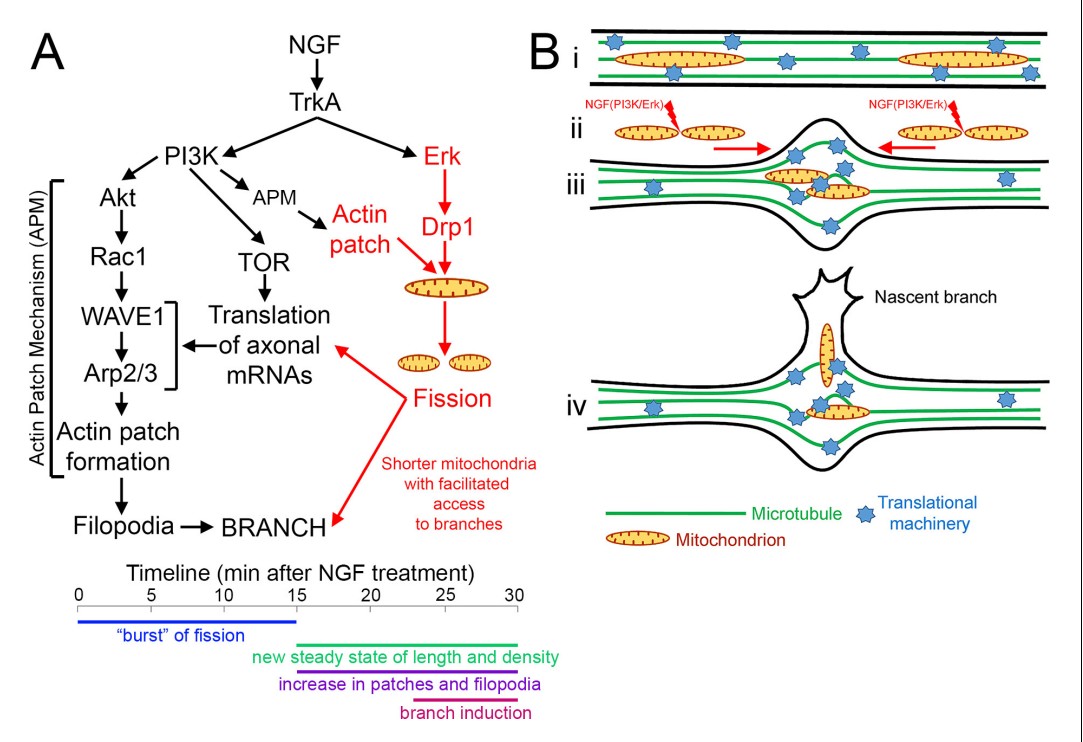

**Figure 8.** Summary and hypothetical mechanism for the role of fission in allowing mitochondria to target to sites of NGF-induced sensory axon collateral branching. (**A**) Schematic summary of previously established components of the mechanism of NGF-induced axon branching (black text; modified from *Spillane et al., 2012*) and the role of NGF-induced mitochondria fission elucidated herein (red text). The formation of actin patches is dependent on a PI3K-Rac1-Arp2/3 based signaling and cytoskeletal mechanism (actin patch mechanism, APM). This mechanism driving formation of actin patches is dependent on PI3K-Akt signaling both in the context of NGF-signaling and in the absence of NGF (*Ketschek and Gallo, 2010*). NGF increases the rate of actin patch formation, in part, by driving the intra-axonal translation of relevant actin regulatory molecules (cortactin, Arp2, WAVE1; *Spillane et al., 2012*) required for patch formation. Herein, we show that NGF also induces the fission of axonal mitochondria during the first 10 min of signaling, prior to branching and the axonal translation-dependent increase in actin patch formation. The fission requires NGF-induced activation of Drp1 through Erk signaling. The fission also requires actin filaments that present as patches at the site of fission. Based on the requirement for actin filaments, PI3K-Akt signaling and the Arp2/3 complex for the formation of actin patches and fission, we suggest the role of PI3K in fission is mediated through the previously described APM also regulating the actin patch component of the fission machinery. The ensuing fission results in mitochondria with lengths that facilitate their targeting into nascent branches. The fission is also required for the NGF-induced and mitochondria-dependent intra-axonal translation of cortactin (*Spillane et al., 2013*). The hypothetical mechanism by which fission may contribute to axonal translation is presented in the panel B. The approximate time line of the events described above is presented below. The NGF-induced burst of fission occurs during the first 10–15 min of signaling, after which the new steady state of length and density is established along axons. The fission precedes the subsequent axonal translation dependent increase in the formation of actin patches and filopodia (*Ketschek and Gallo, 2010*; *Spillane et al., 2012*; *Spillane et al., 2013*). The emerge of new branches from the axon follows becoming evident by after 20 min of NGF treatment (*Spillane et al., 2012*). (**B**) Graphical representation of the hypothetical contribution of fission to the establishment of localized sites of intra-axonal protein synthesis during NGF-induced axon collateral branching. This hypothetical model is derived from the synthesis of from multiple prior studies showing the accumulation of organelles and translational machinery at sites of branching (*Yu et al., 1994*; *Spillane et al., 2013*; *Courchet et al., 2013*; *Wong et al., 2017*), the correlation of sites of branching with localized microtubule splaying (*Dent and Kalil, 2001*; *Ketschek et al., 2015*) and the observation that NGF induces microtubule splaying by 5 min post treatment (*Ketschek et al., 2015*), the time when mitochondria are undergoing fission and redistribution within the axon. (i) before NGF treatment mitochondria are longer, and translational machinery is undergoing transport and stalling throughout the axon in an unregulated manner. (ii) Signaling through PI3K and Erk is required for the fission of axonal mitochondria in response to NGF, and the fission correlates with transport mediated redistribution of mitochondria within the axon. (iii) as mitochondria undergo redistribution they encountered segments of the axon within which microtubules have undergone splaying apart and correlate with sites of potential branching. Through an unspecified mechanism transport cargoes undergo stalling in the axon segments where microtubules have splayed and accumulate resulting in the mitochondria and translational machinery. This converge then assists in the establishment of axon segments with high translation potential that correlate with sites of potential axon branching. (iv) Axon branches can then arise from these specialized domains through a multifaceted process involving the regulation of the actin cytoskeleton, microtubules, intra-axonal protein synthesis and directed transport into branches. The shorter length of mitochondria following fission is permissive for the entry of mitochondria into nascent branches.

for the population of patches that form associated with mitochondria there are two distinct subpopulations with dedicated roles in the formation of filopodia and fission, respectively.

The concentration profile of NGF induced fission on axonal mitochondria is consistent with activation of the TrkA receptor (*Kaplan et al., 1991*) that activates both PI3K and Erk signaling. The observations that inhibition of TrkA using k252a inhibits NGF-induced fission and that treatment with BDNF at a concentration expected to activate the p75 receptor does not induce fission indicate that TrkA is the major receptor mediating the effect of NGF on mitochondria fission. However, a role for p75 in modulating TrkA signaling cannot at present be ruled out. A role for PI3K signaling in regulating Drp1-dependent mitochondria fission has not been previously reported. PI3K signaling, unlike Mek-Erk, does not regulate the activation of Drp1 through S616 phosphorylation. However, actin filament patches are required for the formation of Drp1 focal accumulations at sites of fission. Given the previously detailed role of PI3K in establishing local signaling microdomains in axons that determine the localized formation and elaboration of axonal actin patches that in turn colocalize with mitochondria (*Ketschek and Gallo, 2010*), the role of actin patches in fission and the dependence of both fission and actin patch formation on PI3K and Arp2/3, we propose that PI3K controls the actin patch dependent aspect of the fission mechanism (*Figure 8A*), but do not exclude alternative mechanisms. The inhibition of Arp2/3-mediated actin filament nucleation prevented fission, further indicating that the subpopulation of actin patches that control fission is mediated by the previously detailed PI3K-Akt-Arp2/3 mechanism (*Figure 8A*). Other actin regulatory proteins likely also contribute to the actin-dependent fission induced by neurotrophins. For example, the INF2 formin has been shown to mediate ER-mitochondria contact mediated fission (*Korobova et al., 2013*) and mitochondria and the ER both accumulate at sites of axon branching (*Spillane et al., 2013*). In contrast, Mek-Erk controls the activity of Drp1. Drp1 phosphorylation by Erk has precedent in the cancer literature (*Kashatus et al., 2015*; *Serasinghe et al., 2015*), and is here linked to neurotrophin-mediated regulation of mitochondria but not peroxysomes.

NGF-induced fission is required for the ensuing collateral branching, and inhibition of Drp1 in vivo impairs developmental collateral axon branching in the embryonic spinal cord although this branching is not initiated by NGF (*Patel et al., 2000*). Similarly, a recent study reports that inhibition of mitochondria fission through depletion of MFF, one of the Drp1 adaptors on the mitochondrial surface, decreases the terminal/collateral branching of cortical axons in vivo (*Lewis et al., 2018*). The current study provides insights into the contribution of mitochondria fission through the regulation of mitochondria length and density to the mechanism of branching, independent of the specific signal initiating the branching. The data reveal a 'size principle' for the targeting of mitochondria into nascent branches. In the absence of neurotrophins, only the few available short mitochondria target into spontaneously formed branches. The size of these mitochondria that enter branches in the absence of NGF is similar to the size induced by neurotrophins throughout the axon following fission, indicating that neurotrophins greatly increase the availability of mitochondria in the size range appropriate for branch formation and maturation. *Li et al. (2004)* similarly reported that small mitochondria tend to target to dendritic filopodia and spines, although they also observed portions of larger mitochondria penetrating dendritic protrusions. Furthermore, the data show that when axonal mitochondria undergo fission, *regardless* of NGF treatment, one or both of the emergent mitochondria undergo transport. The increased density of mitochondria in NGF-induced branches is also consistent with increased targeting into nascent branches, as the branches form when NGF has set the new steady state of length and density in axons (*Figure 8A*, see timeline). While the mechanism that links fission with subsequent transport is not clear, an inverse relationship between the length of axonal mitochondria and their propensity for undergoing transport has been reported (*Saxton and Hollenbeck, 2012*; *Narayanareddy et al., 2014*). The length of mitochondria is dependent on the balance of fission and fusion. Therefore, it is also possible that some signals may suppress fusion independent of fission but with the same functional effect in terms of the role of mitochondria length in promoting the targeting of mitochondria to nascent branches.

The temporal aspects of the NGF-induced fission and establishment of the new steady state of length and density relative to the ensuing formation of branches (*Figure 8A*, see timeline), along with consideration of the literature, suggest a hypothetical working model for the role of fission and the subsequent reorganization of mitochondria within the axon in the formation of sensory axon collateral branches induced by NGF (*Figure 8B*). NGF induces a high rate of fission during the first 10–15 min of treatment after which a new steady state of mitochondria length and density is maintained

by NGF signaling. In contrast, the NGF-induced increase in the formation of actin patches and filopodia, and subsequently branches, which are dependent on mitochondria respiration and intra-axonal protein synthesis (*Figure 8A*; *Ketschek and Gallo, 2010*; *Spillane et al., 2012*; *Spillane et al., 2013*; *Sainath et al., 2017a*; *Wong et al., 2017*), become respectively prominent by approximately 15 and 30 min following NGF (*Spillane et al., 2012*). We present the novel observation that instances of fission within the axon correlate with the subsequent transport of one of the emergent mitochondria, indicating that following the initial burst of NGF-induced fission mitochondria also undergo redistribution within the axon, prior to the emergence of branches and the increases in NGF-induced actin patches and filopodia (*Figure 8A*). Branches emerge from sites along the axon where mitochondria have undergone stalling (*Courchet et al., 2013*; *Spillane et al., 2013*; *Tao et al., 2014*). Thus, we suggest that one role of fission is to promote the reorganization of the distribution of axonal mitochondria allowing them the target to sites of future branching. The observation that following NGF treatment the majority of mitochondria runs consist of switches in directionality of movement may represent a mechanism whereby the mitochondrion can repeatedly sample the same axon segment for docking sites. Sites of branching are characterized by localized splaying of the axonal microtubule array (*Dent and Kalil, 2001*; *Ketschek et al., 2015*) and NGF promotes the splaying by 5 min after treatment (*Ketschek et al., 2015*). Thus, as mitochondria are undergoing redistribution within the axon following NGF-induced fission they will encounter sites of microtubule splaying that we suggest may serve to locally capture mitochondria in transit, and lead to the observed accumulation of mitochondria and other organelles at the base of nascent branches (*Yu et al., 1994*; *Courchet et al., 2013*; *Spillane et al., 2013*). Through their respiration stalled mitochondria also establish sites of localized high axonal mRNA translation that correlate with sites of axon branching and are required for the ensuing branching (*Spillane et al., 2013*). Sites of axon branching have also been shown to accumulate ribosomal RNA (*Spillane et al., 2013*). Furthermore, the orchestration in space and time of the accumulation of mitochondria and translational machinery at sites of axon collateral branching has been demonstrated in vivo along retinal ganglion cell axons (*Wong et al., 2017*) whose collateral branches are under regulation by BDNF (*Cohen-Cory et al., 2010*). The study by *Wong et al. (2017)* determined that both mitochondria and translational machinery stall at specific sites along axons supporting the idea that axons have specific sites that capture the relevant machinery (e.g., possibly sites marked by microtubule splaying). Thus, the fission may be required for axonal mitochondria to become redistributed within the axon and targeted to sites wherein additional components of axonal translational machinery are also targeted such as sites of axon collateral branching (e.g., ribosomes, mRNA-protein particles; *Spillane et al., 2013*; *Wong et al., 2017*; *Figure 8B*), a complex issue that will be the focus of future investigation. This hypothetical mechanism (*Figure 8B*) suggesting that mitochondrial fission contributes to a reorganization of the axonal cytoplasm may underlie the observed effects of blocking fission on the axonal translation of cortactin, and these complex issues are now under investigation.

Whether the proposed model may apply to the collateral branching of other neuronal types or in response to other branch inducing signals will have to be addressed by future investigations. The observations that both the developmental collateral branching of sensory axons in the spinal cord and cortical axons in cortex (*Lewis et al., 2018*) requires mitochondria fission, both contexts not considered to be mediated by neurotrophins, indicates that the requirement for fission and the regulation of axonal mitochondria length and density for axon collateral branching is likely to extended to additional collateral branching signals and scenarios. However, the current study and proposed model do not address the issue of the branching of the main axon through growth cone bifurcation or collateral branches that arise from sites along the axon at which the growth cone underwent pausing prior to continuing extension (*Szebenyi et al., 1998*). It will be of interest to address mitochondria behavior and dynamics in the context of these additional forms of axon branching. Similarly, netrin-1 induces the branching of cortical axons through the local regulation of axonal calcium levels through a mechanism requiring both CaMKII and MAPK signaling (*Tang and Kalil, 2005*). In dendrites, activity can drive the fission of mitochondria through a calcium and CamKII-dependent mechanism requiring phosphorylation of Drp1 at S616 (*Divakaruni et al., 2018*). The induction of collateral branches by NGF is independent of calcium signaling (*Gallo and Letourneau, 1998*) and in unpublished studies we have not detected increases in cytosolic calcium or mitochondrial calcium in response to NGF and pharmacological inhibition of CaMKII did not affect NGF-induced branching. The data presented herein are consistent with Mek-Erk mediated phosphorylation of Drp1 at

S616 downstream of TrkA activation. Interestingly, netrin-1 induced branching along cortical axons is dependent on both CaMKII and MAPK signaling (*Tang and Kalil, 2005*), but whether fission of mitochondria downstream of either of these kinases is involved remains to be determined.

Based on our prior work and that of others addressing the role of mitochondria in axon collateral branching (*Courchet et al., 2013*; *Spillane et al., 2013*; *Tao et al., 2014*), we consider that the mitochondrion, its fission and targeting to sites of axon branching are best considered as establishing domains along the axon that are *permissive* for the emergence of branches. As detailed in multiple reviews on the topic of axon collateral branching, the mechanism underlying collateral branching is complex and involves multiple molecular systems controlling multiple aspects of the actin filament and microtubule cytoskeleton (*Kalil and Dent, 2014*; *Pacheco and Gallo, 2016*; *Armijo-Weingart and Gallo, 2017*). Mitochondria positioning along the axon is a major determinant of sites of axonal actin patch formation and filopodia (*Ketschek and Gallo, 2010*; *Spillane et al., 2013*; *Tao et al., 2014*; *Sainath et al., 2017a*), the first step in the mechanism of axon collateral branching. However, in the absence of additional extracellular signals that control cytoskeletal dynamics and rearrangements the mere presence of mitochondria is not sufficient to elicit high levels of actin patch and filopodia formation and in turn collateral branching. Indeed, as we have previously reported about 15% of the axon shaft of the sensory neuron population studied herein contains mitochondria regardless of NGF treatment (*Ketschek and Gallo, 2010*) and branch formation is only increased by NGF treatment, which activates the additional cytoskeletal rearrangements required for formation of branches that however occur preferentially at sites populated by mitochondria (*Figure 8A*). In conclusion, the mitochondrion and its positioning along the axon is a required component of the mechanism of axon branching but we suggest it should not be considered sufficient to elicit axon branches as the promotion of branching by NGF involves both signaling through a canonical cytoskeletal regulatory pathway (PI3K-Arp2/3; *Figure 8A*) and the axonal translation of actin regulatory proteins (*Figure 8A*) and also the promotion of microtubule targeting into nascent branches (*Spillane et al., 2012*).

The mechanism of the maintenance of the neurotrophin-induced steady state of mitochondria in axons will require further consideration. The persistent elevation in Erk and PI3K activity after the initial burst of NGF-induced signaling serve to maintain the steady state, as indicated by the observation of increased mitochondria length when either pathway is inhibited in the continuous presence of NGF. Interestingly, NGF withdraw after the prior establishment of a steady state results in increased rates of fusion but does not impact fission during the period soon after NGF withdraw (10–20 min). Subsequently, fusion rates return to baseline levels and match those of fission, as required for maintenance of the post-NGF withdraw steady state. The mechanism underlying the increase in fusion remains to be determined but may reflect a suppression of PI3K and/or Erk signaling following NGF-withdraw.

In conclusion, this report details that neurotrophins induce an initial burst of mitochondria fission and subsequently maintain a new steady state of mitochondria length and density within axons, and that the regulation of fission is a required component of the mechanism of NGF-induced axon branching. These observations pave the way for considering the role of the effects of neurotrophins on mitochondria fission in the context of the various other functions of neurotrophins spanning the regulation of cell survival, axon elongation/regeneration and synaptic function.

## Materials and methods

### Culturing of primary neurons

Chicken dorsal root ganglia (DRG) were dissected at embryonic day 7 (E7) (SPF eggs obtained from Charles River Laboratories). At this developmental stage it is not possible to determine sex and both sexes were used at presumably a 50/50 ratio. Whole explants were cultured (3–4 explants/dish), or the DRGs neurons were dissociated (as explained below) (1.5 explants/dish). The culturing substrata were previously coated with polylysine (Sigma; Catalog number (Cat#) P9011; 100 µg/ml in borate buffer), for 4 hr and following 3x washing with phosphate buffered saline (PBS) with 25 µg/ml laminin (Life Technologies Cat# 23017–015) in PBS overnight, all incubations were performed at 39°C. Explants and dissociated neurons were cultured in defined F12H medium (Gibco; Cat#21700075). Unless otherwise noted, explants or dissociated neurons were grown in absence of NGF and used

for experiments between 20 and 30 hr after plating. For live imaging experiments, explants or dissociated neurons were plated in glass-bottom dishes. For experiments in which the cultures were fixed, cells or explants were cultured on glass coverslips.

For the preparation of dissociated neurons sensory ganglia were incubated in in $Ca^{2+}$–$Mg^{2+}$-free PBS (CMF-PBS), for 10 min at 37˚C. Ganglia were then spun down for 1 min, and supernatant was removed. Ganglia were then treated with 0.25% trypsin (Fisher Scientific Cat# MT25005CI), for 10 min at 37˚C and spun down for 1 min. Ganglia were then pipette triturated 30 times in F12HS10 media (F12H medium supplemented with 10% fetal bovine serum: Fisher; Cat#MT350111CV) and then spun down for 4 min. Supernatant was removed, cells were triturated again in F12H media 15 times and finally brought to the required volume resulting in dissociated cells that were then plated.

The antibodies to phosphorylated Drp1 did not recognize chicken Drp1 due to amino acid variations within the antigenic site, as supported by inter-species sequence comparison (not shown). Therefore, for studies addressing the effects of NGF on levels of phosphorylated Drp1 we had to use dissociated mouse DRG neurons as mouse Drp1 is recognized by the antibody. P0-P1 mice (C57BL/6; Jackson Laboratories) were placed on ice for 10 min. 1.5 DRGs were dissected per coverslip and placed in CMF-PBS in 37˚C for 15 min. The ganglia were then spun down for 1 min, supernatant was removed and replaced with trypsin, and incubated at 37˚C. After 10 min, the pellet was spun down for 1 min, supernatant was removed and cells were re-suspended in 5 ml F12HS10. Cells were centrifuged for 5 min, supernatant was removed and cells were re-suspended in culturing media containing F12H + NGF. 500 µl of culturing media containing the dissociated cells was added to each coverslip.

## Transfection

For transfection of plasmids into neurons, 40 chicken DRGs were dissociated as described above, and after F12HS10 was removed neurons were suspended in 100 µl nucleofector solution (Lonza Cat# VPG-1002) and gently resuspended through trituration. The DRG cell suspension was transferred to a nucleofector cuvette containing 10 µg of Plasmid DNA, and electroporated using an Amaxa Nucleoporator (program G-13). The electroporated solution was then immediately transferred to a tube containing F12H media as described above prior to plating.

## Immunocytochemistry

For cortactin detection, whole DRGs explants where fixed with 4% paraformaldehyde (Electron Microscopy Sciences Cat# 15710) and 5% sucrose (Fisher Scientific Cat# S5-500) in PBS for 15 min at room temperature and blocked for 30 min with 10% Goat Serum (Sigma Cat# G9023) in PBS with 0.1% of Triton X-100 (Sigma Cat# 9002-93-1) (GST). Samples were incubated for 45 min with primary antibody against cortactin (Abcam, Cat# AB11065) diluted in GST (1/250) at room temperature, and washed with PBS. Next, fluorescently labeled secondary antibody (goat-anti-rabbit TRITC, 1/200; Sigma Cat# T6778) and phalloidin Alexa 488 (1/20, Invitrogen Cat# A12379), were applied for an additional 45 min, at room temperature. After secondary antibody incubation samples were washed with PBS and with deionized water, and mounted with Vectashield mounting (Vector Cat# H-1000). For phosphorylated Akt (Thr308; Cell Signaling Technology, Cat# 9275) immunostaining, the same protocol as explained above was followed using dissociated DRG cultures. For phosphorylated Erk detection dissociated neurons were fixed with methanol for 15 min on ice, and then blocked with 10% Goat Serum in PBS. For the immunostaining the same protocol as explained above was followed. Primary antibody against phosphorylated Erk1/2 (Thr202/Tyr204, 1/500; Cell Signaling Technologies, Cat# 9101), fluorescently labeled secondary antibody (goat-anti-rabbit TRITC, 1:400), and anti-α-tubulin DM1A-FITC (1/100; Sigma, Cat# F2168) was used to counter stain.

In the assessment of axon branches, to detect microtubule polymer, cultures were simultaneously fixed and extracted using 0.2% glutaraldehyde (Electron Microscopy Sciences Cat# 16300) and 0.1% triton X-100 in PHEM buffer (10 mM MES, 138 mM KCl, 3 mM MgCl, 2 mM EGTA), followed by a 15 min incubation in 1 mg/ml sodium borohydride (Fisher Scientific Cat# Ac189301000) in CMF-PBS (*Gallo and Letourneau, 1998*). Samples were then stained with DM1A-FITC anti-α-tubulin to visualize microtubules and rhodamine phalloidin to visualize actin filaments as previously described above.

## Function blocking NGF antibody

For the experiments in *Figure 1F* we used sheep-anti-NGF (CedarLane Labs; Cat# CLMCNET-031). The antibody was added to the medium at 25 µg/mL.

## Labeling of mitochondria

For labeling mitochondria to determine rates of fission and fusion dissociated neurons were transfected with pDsRed2-Mito (Clontech Cat# 632421) as explained above. Mitotracker dyes were prepared according to the manufacturer's directions and labeling was performed through incubation for 30 min with mitotracker green (40 nM, Molecular Probes, Cat# M7514) or red (0.05 nM, Invitrogen, Cat# M22425). Mitotracker red was used for collecting the timelapse data addressing mitochondria movement in *Figure 1—figure supplement 2A–D* due to its greater photostability. Following labeling, the dye containing medium was removed and cultures were washed three times with culturing medium not containing any dye. Imaging was performed at least 30 min after the dye was removed to allow cultures to acclimate.

## Live imaging

Neurons were imaged using a Zeiss 200 M microscope equipped with an Orca-ER camera (Hamamatsu) in series with a PC workstation running Zeiss Axiovision software for image acquisition and analysis. Cultures were placed on a heated microscope stage (Zeiss temperable insert P with objective heater) for 10 min at a constant 39°C before and during imaging. Imaging was performed using a Zeiss Pan-Neofluar 100_objective (1.3 N.A.), 2 × 2 camera binning and minimal light exposure. For the quantification of mitochondria transport and percentage of fission, mitochondria were labeled with mitotracker red and imaged immediately after NGF or storage buffer (control) addition. Acquisition rate was 110 frames with 3 s interframe intervals. For live imaging of transfected neurons 150 frames with 3 s interframe was used. For Drp1 puncta accumulation and percentage of mitochondria quantification, cells were co-transfected with pDsRed2-Mito and pEYFP-C1-Drp1. For analysis of actin patches localizing at mitochondria and fission sites, neurons were co-transfected with eYFP-B-Actin (Clontech Cat# 6902–1) and pDsRed2-Mito. Images were taken immediately after NGF addition. For actin patch formation quantification, cells were transfected with eYFP-B-Actin. Images were taken after 30 min of incubation with NGF. Details of actin patch quantification and imaging are described in *Loudon et al. (2006)*.

## Microperfusion

Using the culturing system and imaging described above we set up a localized microperfusion system to locally deliver NGF to distal axons. The perfusion system was set up as described in *Guijarro et al. (2012)* with modifications. Glass electrodes were pulled to have tips in the 3–5 µm range and filled with NGF containing medium or control medium just prior to use. We included 50 µg/mL NGF and 25 µM CellTracker Green (Thermo Fisher Scientific, Cat# C2925), to visualize the perfusion, in the NGF delivery medium, but only CellTracker Green in the control medium. Pipette positioning and manipulation was performed using a M3301R micromanipulator (World Precision Instruments). To control flow from the pipette we used a Picospritzer III (Parker Hannifin). The pipette tip was positioned approximately 60–80 µm away from and perpendicular to the target distal axon. Delivery was commenced using 5 psi and 20 msec pulses at a frequency of 2 Hz. Flow of medium in the direction parallel to the ejection from the pipette tip, to prevent local accumulation of the perfusion solution, was established by gravity flow using a Kimwipe (KimTech) placed at the opposite end of the culturing dish and draining into a separate dish. Dishes were filled with 4 mL of medium at the start of the experiment and medium was added during the duration of the experiment at the side of the dish opposite the direction of flow to maintain a steady level of medium.

## Microfluidics

Microfluidic chambers were generated in house as described in *Sainath et al. (2017b)* (see *Figure 1—figure supplement 1I* for an example). E6 dissociated chicken DRG neurons were plated within the microfluidic chambers on polylysine and laminin coated coverslips. The cell body compartment media did not contain NGF but was supplemented with FDU ( 5 uM 5-Fluorouracil, Tocris Cat# 325; 5 uM Uridine, Sigma-Aldrich Cat# U3003-5G) to reduce glial cell number. The distal axonal

compartment contained NGF (20 ng/mL) or no NGF. Each cell body chamber contained approximately eight dissociated DRGs and were cultured for 48 hr with additional media added at 24 hr.

Mitochondria in the axons having entered to axon compartment were visualized using Mitotracker green (25 nM, ThermoFisher. The media from the axonal side reservoirs was replaced with media containing Mitotracker green in one of the axonal compartment reservoirs such that it flowed across the axons to reach the opposite side axonal reservoir. Axons were incubated in Mitotracker for 20 min followed by two washes with dye free medium before imaging at 100x. To wash out the Mitotracker dye, the media was removed from both axonal reservoirs and dye free media was added to one side of the axonal reservoir and allowed to flow over the axons to the opposite axonal reservoir.

## Quantification of immunocytochemical labeling

For quantification of the levels of immunolabeling in axons the distal 50 µm of axons were sampled excluding the growth cone, defined as the region of the axon distal to the 'neck' of the growth cone where the axon diameter enlarges into the central and peripheral domains. Images were acquired using a Zeiss 200 M microscope with a camera binning of 2 × 2. For imaging a Zeiss Pan-Neofluar 100 objective with 1.3 NA, was used. The exposure parameters were set so as to maintain all images within the dynamic range without any pixels at saturation. Prior acquisition of imaging data sets, exposure parameters were determined from the control groups so that the mean axonal intensity of the control group was at about 30% of the dynamic range, providing for a two-fold increase in experimental groups that would still remain with the range. Zeiss Axiovision software was used for all measurements, and the area of the region of interest was multiplied by the background-subtracted mean intensity of the stain to obtain the total integrated level of fluorescence.

## Determination of mitochondria number, length, fission and fusion

Imaging was performed as described in the live imaging section above but camera binning was set to 1 × 1 to obtain maximal spatial resolution. The length and number of mitochondria in the axon were measured using ImageJ software, and the density of mitochondria was calculated per 10 µm of axon. Mitochondria length was defined as determined by a line, segmented if the mitochondrion was not strictly linear, running from one end to the other of the mitochondrion. Occasionally, mitochondria in axons overlapped as clearly evidenced by the additive signal of their fluorescence in the region of overlap. Overlapping mitochondria were excluded from analysis to ensue only mitochondria with clearly defined ends were used for quantification.

## Determination of rates of fission and fusion from live imaging time lapse videos

Instances of fission were determined manually by the user through frame by frame analysis using sequences as that shown in *Figures 1A* and *2A* and *Figure 1—figure supplement 1*. The labeling of mitochondria using both dyes and the genetically encoded reporters was uniform along the mitochondrion, although with DsRed-mito mitochondria often exhibited differences in mean fluorescence intensities. Using the settings described in the live imaging section each pixel in the resulting images has a radius of 0.12 µm. Examples of fission and fusion and the relevant analytical considerations are shown in *Figure 1—figure supplement 1*. A completed fission event was defined as when the fluorescence intensity of the mitochondrion at the site of apparent fission had decreased to the background level of the axonal segment not containing mitochondria and the two resultant mitochondria were separated by over 5 pixels representing 0.6 µm (*Figure 1—figure supplement 1*). Occasionally mitochondria underwent apparent constrictions but did not continue into a full fission event, as the constriction sites returned to exhibit the prior width and intensity, and these instances were not counted as fission. Fusion events were similarly determined by the user. When mitochondria overlap in space the intensity signal is additive (*Figure 1—figure supplement 1*), allowing discrimination of overlap from fusion. Fusion was thus determined to occur in instances when mitochondria ends came into contact and then formed an apparent single mitochondrion with uniform intensity, verified by line scans using NIH ImageJ (*Figure 1—figure supplement 1*). As noted above, DsRed-mito often labeled mitochondria resulting in different mean intensities per mitochondrion. The redistribution of signal intensity from the higher intensity mitochondrion to equalize within the emergent fused mitochondrion was also taken as a criterion of fusion when disparities between the intensity of the

two interacting mitochondria were evident (*Figure 1—figure supplement 1*). Also, following instances considered to reflect fusion the lengths of the considered emergent mitochondrion was additive of that of the two contributing mitochondria (*Figure 1—figure supplement 1*). A caveat is that we cannot exclude the possibility of rare events during which two mitochondria ends might have come into immediate proximity (e.g., less than 0.24 µm/2 pixels) giving the appearance of fusion, as defined by the apparent uniform intensity of the resulting dyad of adjacent mitochondria. However, as measured, the fusion rates were not affected in any case in this study and this caveat is unlikely to have biased results.

## Analysis of mitochondrial movement

The movement of mitochondria, as with fission/fusion dynamics, was determined through detailed frame-by-frame analysis. Stalled mitochondria were defined as those that spent at a minimum the first minute during the imaging sequence without detectable movement. Approximately, 60% of mitochondria thus defined as stalled, that did not undergo fission, remained stalled for the duration of the imaging period. The remainder provided the baseline for stalled mitochondria that did not undergo fission but that may have subsequently undergone transport. Oscillating mitochondria were defined as those that underwent bouts of antero-retrograde movements less that 1 µm in either direction but did not undergo net displacement during the imaging period, analogous to the 'dynamic pause' behavior described by *Chen et al. (2016)*. Mitochondria undergoing transport were defined as those having moved greater than 1 µm in the retrograde or anterograde direction. Rates of mitochondrial transport (µm/sec) were derived by measuring the displacement of the mitochondria end in the direction of transport during the period of continuous movement in that direction divided by the time during which the movement occurred. If a retrogradely transported mitochondrion exited the field of view the measurement was performed up to the last frame when the mitochondrial end was visible, and if an anterograde mitochondrion entered the growth cone the measurement was similarly ended. Measurements were made along the axon shaft up to the neck of the growth cone, where the axon shaft increases in diameter as the central domain commences, or if a distinct neck was not visible, as is the case with growth cone that do not have lamellipodia, then the distal 10 µm of the axon shaft were defined as the growth cone and excluded. The lengths of runs were determined by the sum of the distance a mitochondrion moved during a continuous bout of movement. If a mitochondrion switched direction during a run the total run length is reflective of the summed distance traveled during the run regardless of direction. Termination of movement during a run was defined as a period greater than or equal to 6 s during which no movement occurred.

## Quantification of axon branching

The criteria for determination of mature branches from in vitro experiments wherein the microtubules and actin filaments were labeled as described above her been extensively detailed in *Spillane et al. (2012)*, and *Spillane et al. (2013)*, and were followed herein. Briefly, a branch was defined as containing microtubules and an asymmetric distal enriched accumulation of actin filaments. Similarly, for determination of branches in the spinal cord we followed the morphological criteria described in *Spillane et al. (2011)*. Briefly, extensions along the axon shaft greater than 10 µm were classified as branches.

## Axonal actin dynamics

The methods for the analysis of the dynamics of axonal actin patches have been described in detail in *Loudon et al. (2006)* and were similarly followed in this work. Given that Drp1 accumulations followed similar dynamics to actin patches the same approach was used to analyze the former. As described for the quantification of immunocytochemical staining levels acquisition parameters were set so as to keep the pixels in the image within the dynamic range and $2 \times 2$ camera binning was used.

## Fluorescence recovery after photobleaching (FRAP) Analysis

For FRAP experiments dissociated neurons were electroporated with myrGFP3'UTR-Cortactin (*Spillane et al., 2012*). As described for the quantification of immunocytochemical staining levels acquisition parameters were set so as to keep the pixels in the image within the dynamic range and 2 × 2 camera binning was used. Images were acquired immediately after NGF addition, and just before and after bleaching, at 5 min intervals for 1 hr. For imaging a Zeiss Pan-Neofluar 100 objective with 1.3 NA was used. Photobleaching of myrGFP was performed using a 2 min exposure of the distal axon in the imaging field to light from a 100 W source. Phase contrast images of the axons acquired at the same times as the GFP channel were used to determine the location of the axon at early time points in the FRAP when the GFP signal is too low to detect the axon. Only axons that were photobleached by 90% of their initial value were used in the analysis. To track recovery, the mean background subtracted intensity of myrGFP was measured in a fixed length of the axon at multiple time points.

## In ovo electroporation and ex vivo imaging

As described in *Spillane et al. (2011)*, *Spillane et al. (2012)*, *Spillane et al. (2013)* and *Donnelly et al. (2013)*, lumbosacral chicken embryo DRG were electroporated in ovo at E3 using a CUY-21SC electroporator (Nepa Gene) equipped with 3 mm L-shaped gold tip electrodes (Harvard Apparatus). Five 50 ms 50V pulses were applied at a rate of 1 pulse per second. Expression vectors were injected (1 µg/µl diluted in dH2O with 0.01% fast green) into the lumen of the neural tube and electrodes placed at the level of the lumbosacral enlargement. DRGs were transfected unilaterally. At E7, embryos were removed from the eggs and the entire spinal cord caudal to the first thoracic segment was dissected. The cord was then divided into two halves by cutting the roof and floor plates and immediately placed in the well of a video-imaging dish containing 20 µl of culturing medium. The well of the video dish was then closed by placing a glass coverslip on top of the well in the dish and sealed to the dish by a ring of medium between the glass and dish surface. The assembled video dish containing the spinal cord was immediately used for imaging. Imaging was performed using a Zeiss 200M inverted microscope equipped with an Orca-ER CCD camera (Hamamatsu). For imaging a Zeiss Pan-Neofluar 100X objective with 1.3 NA was used (*Spillane et al., 2012*). Plasmid used for in ovo electroporation were: pDsRed2-Mito, pEYFP-C1-Drp1, mCherry, pEYFP-C1-Drp1K38A.

## Generation of dominant negative K38A Drp1

The plasmid pEYFP-C1-Drp1 (Addgene; Cat# 45160) was used as the PCR template of directed mutagenesis (Agilent Technologies; Cat# 200519–4) to create the mutant plasmid pEYFP-C1-Drp1K38A. The AAG (K38) codon of *H. sapiens* DRP1 (GeneBank ID: NM_005690) was mutated to GCG (A) using the following primer sequences: nucleotides 259–295, sense-GAACGCAGAG-CAGCGGAGCGAGCTCAGTGCTAGAAAG and antisense-CTTTCTAGCACTGAGCTCGCTCCGC TGCTCTGCGTTC. The PCR conditions were set following the manufacture instructions as 95℃ for 30 s, 55℃ for 1 min, and 68℃ for 7 min in a duration of 16 cycles. The PCR product was fully verified by sequencing.

## Quantification and statistical analysis

In vitro data involving population based statistical comparisons (i.e., axon branching, quantitative immunocytochemistry, mitochondria length and density) was collected from 4 to 6 cultures per experimental group sampling only axons that were not containing other cells along the regions where the analysis was performed. Individual cultures were generated using explants or cells from different embryos (herein defined as a biological replicate). All groups in the experimental design were cultured, treated and processed in parallel. For live imaging studies, the number of cultures is reflected in the number of axons sampled. Sampling between experimental groups within an experimental design was equalized across days so that data for different groups did not come from disparate sets of cultures sampled on different days. For in ovo electroporation studies 3–4 chicken embryos were electroporated and axons sampled therein, each chicken embryo is considered a biological replicate. Quantification was performed blind after masking data sets and identification of the groups from a masking key upon completion of the analysis.

All data were analyzed using Instat software (GraphPad software Inc). The software determines the normalcy of data sets using the Kolmogorov and Smirnov test. For normal data sets analysis was performed using the Welch t-test for independent groups or the paired t-test for before-after treatment experimental designs. If non-normal data sets were detected, then non-parametric analysis was used (Mann-Whitney test). For multiple comparison tests within experimental designs, parametric Bonferroni or non-parametric Dunn's post-hoc tests were used based on the normalcy of the data sets. For data sets representing categorical data falling into bins the Fischer's exact test was used on the raw data, although the data is expressed as percentages for ease of appreciation. One or two tailed p values are reported based on whether the hypothesis did or did not dictate the directionality of the expected change in mean or median, respectively. All graphs were generated using Prism (GraphPad software Inc). Sample sizes and qualitative statistical presentation are denoted in figure legends or figures.

## Pharmacological reagents

LY294002 (Sigma, Cat# L9908), cycloheximide (Sigma, Cat# 01810), U0216 (Sigma, Cat# U120), PD325901 (Sigma, Cat# 444968), mDivi-1 (Sigma, Cat# MO1999).

## Acknowledgements

This work was supported by NIH/NINDS (NS095471) to GG and GS and NIH/NINDS (NS078030) to GG We thank Dr. D Mochly-Rosen (Stanford University) for the gift of the P110 peptide and Dr. G Thomas (Temple University) for the gift of the function blocking anti-NGF antibody.

## Additional information

### Funding

| Funder | Grant reference number | Author |
|---|---|---|
| National Institute of Neurological Disorders and Stroke | R01NS095471 | George M Smith Gianluca Gallo |
| National Institute of Neurological Disorders and Stroke | R01NS078030 | Gianluca Gallo |

The funders had no role in study design, data collection and interpretation, or the decision to submit the work for publication.

### Author contributions

Lorena Armijo-Weingart, Conceptualization, Formal analysis, Investigation, Methodology, Writing—original draft; Andrea Ketschek, Conceptualization, Investigation, Methodology; Rajiv Sainath, Almudena Pacheco, Investigation, Methodology; George M Smith, Conceptualization, Funding acquisition, Methodology, Project administration; Gianluca Gallo, Conceptualization, Resources, Data curation, Formal analysis, Supervision, Funding acquisition, Investigation, Methodology, Writing—original draft, Project administration

### Author ORCIDs

Gianluca Gallo (iD) https://orcid.org/0000-0002-0777-379X

### Decision letter and Author response

Decision letter https://doi.org/10.7554/eLife.49494.sa1
Author response https://doi.org/10.7554/eLife.49494.sa2

## Additional files

### Supplementary files

• Transparent reporting form

## Data availability

Data have been deposited in Dryad Digital Repository (https://doi.org/10.5061/dryad.5tb2rbp0v).

The following dataset was generated:

| Author(s) | Year | Dataset title | Dataset URL | Database and Identifier |
|---|---|---|---|---|
| Lorena Armijo-Weingart, Andrea Ketschek, Rajiv Sainath, Almudena Pacheco, George M Smith, Gianluca Gallo | 2019 | Data from: Neurotrophins Induce Fission of Mitochondria Along Embryonic Sensory Axons | https://doi.org/10.5061/dryad.5tb2rbp0v | Dryad, 10.5061/dryad.5tb2rbp0v |

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
