## [Decision Letter]

**Acceptance summary:**

This study reports that NGF induces mitochondrial fission in sensory neurons, resulting in collateral axon branching. The results confirm that neurotrophic signaling occurs through mitochondrial fission, which has been actively investigated recently by several groups. The impact of these studies is the suggestion that effects on morphological plasticity by neurotrophins depends upon mitochondrial function.

**Decision letter after peer review:**

Thank you for submitting your article "Neurotrophins Induce Fission of Mitochondria Along Embryonic Sensory axons" for consideration by *eLife*. Your article has been reviewed by three peer reviewers, and the evaluation has been overseen by a Reviewing Editor and Marianne Bronner as the Senior Editor. The following individual involved in review of your submission has agreed to reveal their identity: Stefan Kassabov (Reviewer #2).

The reviewers have discussed the reviews with one another and the Reviewing Editor has drafted this decision to help you prepare a revised submission.

This study shows that NGF controls mitochondria fission which gives rise to axon branching in sensory neurons. The findings are novel and have a functional readout. There was considerable agreement that the study was well done and of interest. A number of major reservations were raised. The major issues are the need for compartmentalized cultures, if possible, to confirm the local effects of NGF and more pharmacological controls for mitochondria fission and axon branching. The controls include blocking Trk and PI3K and Erk pathways and are detailed in the reviews below. Several corrections in the text and additional discussion of the literature on mitochondrial fission are recommended. The reviewers also raise several pertinent questions and issues that should be addressed.

Reviewer #1:

The authors report that neurotrophins induce mitochondrial fission in the embryonic chick sensory axons, which is followed by a steady state with stable mitochondria length and density. The NGF dependent branching is driven by combined PI3K and Mek-Erk signaling. Mek-Erk controls the activity of the fission mediator Drp1 GTPase, while PI3K may contribute to the actin dependent aspect of fission. It is interesting that the Drp1 mediated fission is specific to mitochondria and the regulation of mitochondrial fission is a required component of the NGF-induced axon branching mechanism. Overall this manuscript is well written, well referenced and rigorous. I am a bit torn with respect to how transformational this work is. I would argue that the majority of the work is connecting dots that are already out there not only from this group but also folks like Hollenbeck and Koyama. For example it's known that mitochondria are important for branching, Drp1 is required for fission, and that MAPK phosphorylates Drp1 among other things. The real novelty here is the NGF promotes mitochondrial fission which is required for branching. They fill in a lot of details for how NGF signaling fits into these previously published observations. Below are individual critiques:

Axons were locally perfused axons with NGF to address whether NGF is acting locally to induce mitochondrial fission (Figure 1—figure supplement 2G). It's hard to be totally confident that this is local. Is it possible to do these experiments in scenarios where the cell body is isolated (i.e. compartmentalized cultures)?

If actin patch formation is an NGF dependent process as shown by their previous work (Ketschek and Gallo, 2010), what are the dynamics of mitochondria fission and filopodia formation from the actin patch in presence of NGF? What is the timeline for actin patch formation after mitochondria fission?

Reviewer #2:

In the current manuscript Galo and colleagues uncover a novel function of neurotrophins in regulating mitochondrial fission, which enables colateral branching in embryonic chick sensory axons in vitro. While the role of mitochondrial fission in axonal branching in vitro and in vivo has already been demonstrated in a recent paper by Franck Polleux's group (Lewis et al., 2018), the demonstration that neurotrophins exert their well-recognized effects on axonal branching through regulation of mitochondrial fission is indeed new and significant. It adds to a growing body of evidence connecting neurotrophic signaling with mitochondrial function and implies that other important functions of neurotrophins in neuronal morphogenesis and synaptic plasticity could be similarly dependent on their effect on mitochondrial dynamics.

The study is conducted in a very rigorous and thorough manner and the conclusions of the authors are well supported by the data. The papers is well written and the authors have thoroughly and comprehensively discussed their findings in the context of axon collateral branching. However, the authors should do more to contextualize their findings in a broader perspective of neurotrophin biology and its connection with mitochondrial function. There are some major omissions from their review of the existing literature, which are very pertinent and need to be cited and discussed as pointed below. There are also some additional experiments needed to extend and strengthen the author's conclusions as detailed below.

1) Literature review

Most critically the authors should cite and discuss a recent study by Tom Blanpied's group, that found that NMDAR dependent LTP induction is accompanied by and requires a rapid burst of dendritic mitochondrial fission (Divakaruni et al., 2018). The burst of mitochondrial fission triggered by LTP inducing stimuli is very similar to the one identified by the current study by Gallo's group, which raises the exciting possibility that BDNF release, that is well known to be critically important for LTP induction, is mediating the burst. The Blanpied's group study found that cytosolic Ca^2+^ elevation triggered the mito fission burst in a CaMKII, actin and Drp1 dependent manner. Both studies found that phosphorylation of Drp1 at Ser615 is critical and although Blanpied study concluded that CaMKII is the responsible kinase, it is also possible that the CaMKII role is indirect and needed for the secretion of BDNF and the actual kinase involved is Mek-Erk downstream of BDNF release as shown in the Gallo's study. There are examples from the broader non neuronal literature of both kinases phosphorylating Drp1 so they could also be differentially involved in the two different contexts.

In this context the authors should also cite the seminal study by Morgan Sheng's group (Li et al., 2004) showing that modulation of neuronal activity (KCl depolarization or TTX silencing) bi-directionally controls mitochondrial fission in a Drp1 dependent manner and mitochondrial fission is critical for synaptogenesis. That study found a higher level of invasion of the smaller mitochondria resulting from increased fission into nascent filopodial which are precursors of spines which led to formation of more excitatory synapses. That is very reminiscent of the higher level of invasion of smaller mitochondria into filopodial precursors of branches found in the Gallo study. Moreover, as in the more recent LTP study – BDNF release which is triggered by elevated neuronal activity could well be the factor inducing the fission observed in the earlier study, as the current Gallo study would imply.

The idea that neurotrophin release mediated mitochondrial fission is underlying branching, synaptogenesis and synaptic plasticity is a very exciting proposition, with large clinical ramifications that should be given a proper airing in the paper and will broaden its impact and appeal.

2) Experimental i) One important aspect of the study that has not been sufficiently elaborated, is the mechanism of persistence of the new steady state of mitochondrial fission/fusion balance in the presence of NGF. How is this balance stabilized at a certain set point and maintained and how does it revert back to its pre-existing level upon NGF withdrawal? After the initial burst of increased fission/fusion leading to smaller mitochondrial size, this ratio is brought back to equal to maintain that size. When NGF is withdrawn, either the fission rate must decrease or the fusion rate increase until the pre-existing mito length is achieved. The authors should analyze more closely the mitochondrial dynamics changes upon NGF withdrawal by live imaging at different points after NGF withdrawal to distinguish these possibilities and get a better sense of the kinetics of the reversal process. Moreover, they should block the PI3K-AKT and/or Mek-Erk pathways AFTER the establishing of the steady state to distinguish their relative contributions to the maintenance of this state.

ii) There are multiple studies showing that neurotrophin signaling modulates mitochondrial functions apart from mitochondrial dynamics and Ca^2+^ handling, including ATP synthesis, membrane potential, respiratory capacity and others, some of which are cited by the authors themselves. Moreover, mitochondrial dynamics is intimately linked with mitochondrial functions. Given that it seems likely that some of these functional parameters could also be changed, so in addition to their smaller size the mitochondria in the new NT driven steady state could also be functionally distinct and more active. At minimum, the membrane potential can be relatively straightforwardly assessed using TMRM staining, but a more thorough investigation of other parameters like ATP production and respiratory capacity will be very informative.

iii) The authors have thoroughly analyzed the effects of exogenous NGF applications on mitochondrial fission, but it is important to assess what is the contribution of endogenous NT signaling. The authors can use function blocking antibodies and or pharmacological blockade of NT-Trk signaling to ask if the baseline level of fission/fusion balance is influenced. Given the finding of Blaniped's group that CamKII signaling is dynamically involved in the baseline maintenance of fission/fusion balance in addition to its role in LTP induction in hippocampal neurons, it will be interesting to asses CaMKII's role in the context of the current study.

iv) Since neurotrophins signal through both Trk and P75NTR receptors it is standard practice to assess what is the contribution of each class of receptor to the observed phenomenon. The authors do show that classic Trk mediated pathways are involved, but this does not automatically rule out p75NTR involvement, which can also augment Trk signaling in some contexts or antagonize it in others. This will be worthwhile to be assessed in the maintenance phase as well. As above, function blocking antibodies against Trk and p75NTR and pharmacological means (K252A) can be used to address this question.

Reviewer #3:

The paper entitled "Neurotrophins induce fission of mitochondria along embryonic sensory axons" is describing nicely a novel role of short-term mitochondria fission as a promoter of axonal branching of sensory neurons, in an NGF dependent process. This neurotrophin signaling is mediated through both ERK and AKT pathways, leading to fast localized translation of actin related proteins that has a dual role both in filopodia formation and in mitochondria fission that contribute further to branching.

The manuscript displays a strong, validated system, and using a set of pharmacological and genetical manipulations is providing strong evidence supporting the novel and innovative hypothesized model.

1) For all genetic and pharmacologic manipulations – are they having effect on neuronal survival and cause degeneration? The author is using high levels of drugs such as Lantrunculin that is toxic to neurons but there is no control experiment measuring the effect of those drugs on neuronal viability in the current DRG system.

2) Figure 1—figure supplement 2G-H are discussing the effects of a "local" perfusion of NGF. However, as those experiments were not done under any separation or isolation of specific parts of the axon. The authors should either do those experiments in a compartmental chamber or by using labeled NGF so it can be interpreted as local effect.

3) The IF imaging of Phos-DRP, Phos-ERK and Phos-AKT (Figure 4) are not in high quality, and they are the only direct evidence (not pathway inhibition) used to describe the time course of NGF effect on their neuronal system. The authors should provide experiments with higher resolution and imaging quality. Also, the authors are claiming that P-AKT and P-ERK are higher in mitochondria positive axonal segments. However, the signal enrichment in those parts after 6 minutes is only around 150%, but when the authors measured the whole axon they found an enrichment of 450%, thus contradicting their own statement.

4) The in-vivo data is referring only to the effect of DN-DRP1 on mitochondria length and density, and number of axonal branches. This experiment is discussing a long-term effect of DRP1 inhibition, with no reference to the acute effect of NGF on this system. The experiments of the acute ex-vivo explants should be also tested after administration of NGF to see that NGF administration will not cause a rescue effect, meaning the observed changes are a result of the chronicle toxicity of DN-DRP1 on the embryo.

5) Figure 7 – The local synthesis results are very promising but require many crucial controls.

a) What is the effect of protein synthesis inhibition (puromycin, anisomycin, cycloheximide) on mitochondria fission and axonal branching of the sensory neurons? Also, a protein synthesis inhibition should be used as a control to see the baseline signal of no-recovery after photo-bleaching.

b) What is the effect of AKT or ERK pathway inhibitors on the local translation of cortactin?

c) Is there endogenous translation of cortactin after NGF treatment?

6) It is a long paper with complexed figures and inadequate writing that make it hard for the reader to follow.

---

## [Author Response]

Reviewer #1:[…] Axons were locally perfused axons with NGF to address whether NGF is acting locally to induce mitochondrial fission (Figure 1—figure supplement 2G). It's hard to be totally confident that this is local. Is it possible to do these experiments in scenarios where the cell body is isolated (i.e. compartmentalized cultures)?

In addition to the originally presented data sets involving local microperfusion of distal axons through pipette-mediated delivery of NGF (Figure 1—figure supplement 2G, H) we now also present data from experiments using microfluidic chambers providing evidence that axons in the NGF containing distal compartment exhibit mitochondria of shorter lengths than those growing into a compartment not containing NGF (Figure 1—figure supplement 2I).

“As an alternative approach to locally expose distal axons to NGF we used microfluidic compartmentalized chambers and measured the length of mitochondria in axons that grew into compartments containing no NGF or NGF (the cell body compartments did not contain NGF). Axons in compartments containing NGF exhibited shorter mitochondria relative to no NGF (Figure 1—figure supplement 2I), consistent with a local action of NGF.”

Added to Materials and methods;

“Microfluidics

Microfluidic chambers were generated in house as described in Sainath et al.

(2017b; see Figure 1—figure supplement 2I for an example). […] To wash out the Mitotracker dye, the media was removed from both axonal reservoirs and dye free media was added to one side of the axonal reservoir and allowed to flow over the axons to the opposite axonal reservoir.”

If actin patch formation is an NGF dependent process as shown by their previous work (Ketschek and Gallo, 2010), what are the dynamics of mitochondria fission and filopodia formation from the actin patch in presence of NGF? What is the timeline for actin patch formation after mitochondria fission?

We clarify that patch formation is not “an NGF-dependent process”. NGF increases the rate of actin patch formation, but actin patches also form in the absence of NGF treatment (Ketschek and Gallo, 2010). Actin patches serve as precursors to filopodia formation regardless of NGF treatment and NGF does not elicit increased rates of filopodia formation from patches (i.e., the probability that a patch will give rise to a filopodium; Ketschek and Gallo, 2010) or alter other aspects of patch dynamics (at the population level). We previously described that increases in the rate of actin patch formation become detectable at 15 min post-treatment but reach the highest level at 30 min after treatment (Spillane et al., 2012). The increase in the rate of actin patch formation induced by NGF is dependent on the axonal translation of mRNAs coding for actin regulatory proteins (Spillane et al., 2012) involved in patch formation and development (Spillane et al., 2011. 2012).

What is the timeline for actin patch formation after mitochondria fission?

The issue and the timeline of events (e.g., increase in actin patch formation and fission following NGF treatment) is presented in the Discussion and model figure (Figure 8A). A novel aspect of the current submission is the identification of a population of actin patches that is dedicated to fission and does not give rise to filopodia (Figure 3).

We have modified a statement in the Discussion emphasizing the point made above. “In contrast, the NGF-induced increase in the formation of actin patches and filopodia, and subsequently branches, which are dependent on mitochondria respiration and intra-axonal protein synthesis (Figure 8A; Ketschek and Gallo, 2010; Spillane et al., 2012, 2013; Sainath et al., 2017; Wong et al., 2017), become respectively prominent by approximately 15 and 30 min following NGF (Spillane et al., 2012).”

Reviewer #2:[…] The study is conducted in a very rigorous and thorough manner and the conclusions of the authors are well supported by the data. The papers is well written and the authors have thoroughly and comprehensively discussed their findings in the context of axon collateral branching. However, the authors should do more to contextualize their findings in a broader perspective of neurotrophin biology and its connection with mitochondrial function. There are some major omissions from their review of the existing literature, which are very pertinent and need to be cited and discussed as pointed below. There are also some additional experiments needed to extend and strengthen the author's conclusions as detailed below.1) Literature reviewMost critically the authors should cite and discuss a recent study by Tom Blanpied's group, that found that NMDAR dependent LTP induction is accompanied by and requires a rapid burst of dendritic mitochondrial fission (Divakaruni et al., 2018). The burst of mitochondrial fission triggered by LTP inducing stimuli is very similar to the one identified by the current study by Gallo's group, which raises the exciting possibility that BDNF release, that is well known to be critically important for LTP induction, is mediating the burst. The Blanpied's group study found that cytosolic Ca^2+^ elevation triggered the mito fission burst in a CaMKII, actin and Drp1 dependent manner. Both studies found that phosphorylation of Drp1 at Ser615 is critical and although Blanpied study concluded that CaMKII is the responsible kinase, it is also possible that the CaMKII role is indirect and needed for the secretion of BDNF and the actual kinase involved is Mek-Erk downstream of BDNF release as shown in the Gallo's study. There are examples from the broader non neuronal literature of both kinases phosphorylating Drp1 so they could also be differentially involved in the two different contexts.In this context the authors should also cite the seminal study by Morgan Sheng's group (Li et al., 2004) showing that modulation of neuronal activity (KCl depolarization or TTX silencing) bi-directionally controls mitochondrial fission in a Drp1 dependent manner and mitochondrial fission is critical for synaptogenesis. That study found a higher level of invasion of the smaller mitochondria resulting from increased fission into nascent filopodial which are precursors of spines which led to formation of more excitatory synapses. That is very reminiscent of the higher level of invasion of smaller mitochondria into filopodial precursors of branches found in the Gallo study. Moreover, as in the more recent LTP study – BDNF release which is triggered by elevated neuronal activity could well be the factor inducing the fission observed in the earlier study, as the current Gallo study would imply.The idea that neurotrophin release mediated mitochondrial fission is underlying branching, synaptogenesis and synaptic plasticity is a very exciting proposition, with large clinical ramifications that should be given a proper airing in the paper and will broaden its impact and appeal.

Based on the above discussed literature on mitochondria fission in the context of neuronal activity, the reviewer raises the pertinent question of the possible role of calcium signaling in the mechanism of NGF induced mitochondria fission and branching and asks for discussion of recent literature relevant to this issue of calcium. The mechanism of NGF induced sensory axon branching, unlike that described for some central nervous system neurons (e.g., Netrin induced branching along cortical axons; Tang and Kalil, 2005), does not involve calcium (Gallo and Letourneau, 1998). Over the years, we have further probed possible roles for calcium in unpublished experiments and invariably have failed to observe any role for calcium.

First, using cytosolic ratiometric calcium sensors (GCamp6) we fail to see and effect of NGF on axonal calcium levels. Second, using mitochondrially targeted ratiometric calcium sensors (mito-GCamp6) we also fail to detect an effect of NGF on mitochondrial calcium levels. In the context of these data sets we had positive controls using calcium ionophores. Third, using KN62 to inhibit CaM Kinases we do not observe an inhibition of NGF-induced axon branching (we used a 1-30 µM, 30 min pre-treatment prior to treatment with NGF; KN62 is expected to have saturating inhibitory effects at 5 µM). While we do not feel that these data sets addressing the role of calcium would add to the current manuscript we are now including a discussion of the absence of a role of calcium in NGF-induced axon branching and the literature suggested by the reviewer (Discussion, sixth paragraph). We are also referencing the Li et al., 2004 paper in the context of the discussion regarding mitochondria size and targeting to branches (Discussion, fourth paragraph). The added sections of text are reproduced below:

“Similarly, netrin-1 induces the branching of cortical axons through the local regulation of axonal calcium levels through a mechanism requiring both CaMKII and MAPK signaling (Tang and Kalil, 2005). […] Interestingly, netrin-1 induced branching along cortical axons is dependent on both CaMKII and MAPK signaling (Tang and Kalil, 2005), but whether fission of mitochondria downstream of either of these kinases is involved remains to be determined.”

“Li et al., 2004, similarly reported that small mitochondria tend to target to dendritic filopodia and spines, although they also observed portions of larger mitochondria penetrating dendritic protrusions.”

2) Experimental i) One important aspect of the study that has not been sufficiently elaborated, is the mechanism of persistence of the new steady state of mitochondrial fission/fusion balance in the presence of NGF. How is this balance stabilized at a certain set point and maintained and how does it revert back to its pre-existing level upon NGF withdrawal? After the initial burst of increased fission/fusion leading to smaller mitochondrial size, this ratio is brought back to equal to maintain that size. When NGF is withdrawn, either the fission rate must decrease or the fusion rate increase until the pre-existing mito length is achieved. The authors should analyze more closely the mitochondrial dynamics changes upon NGF withdrawal by live imaging at different points after NGF withdrawal to distinguish these possibilities and get a better sense of the kinetics of the reversal process. Moreover, they should block the PI3K-AKT and/or Mek-Erk pathways AFTER the establishing of the steady state to distinguish their relative contributions to the maintenance of this state.

As suggested by the reviewer, we now present data on the percentage of mitochondria that undergo fission and fusion in the context of the NGF withdraw paradigm initially used in Figure 1F. The new data are presented in Figure 1G.We found that during the first 10-20 minutes after NGF withdraw there is an increase in fusion with no effect on fission. In contrast, during 70-80 minutes after NGF withdraw the rate of fusion returned to baseline levels, and that of fission was not affected. These data thus indicate, somewhat surprisingly, that upon NGF withdraw there is an activation of fusion without an impact on fission. This is an interesting observation and we plan of following it up in future work.

We also addressed the roles of PI3K and Mek-Erk signaling in maintaining mitochondria length during the NGF-induced steady state and find that inhibition of either of these pathways results in mitochondrial elongation (Figure 4—figure supplement 1K).

We had initially considered that the analysis of the mechanism of the mitochondrial steady state maintained by NGF would be a detailed follow up publication. However, based on the reviewer’s suggestions we now present these data herein and agree that they add to the present manuscript. We plan on following up on these data sets in a subsequent manuscript.

We have revised the text in the Discussion regarding the steady state to reflect the new observations, resulting in shortening of that section and removal of the more speculative aspects.

“The mechanism of the maintenance of the neurotrophin-induced steady state of mitochondria in axons will require further consideration. […] The mechanism underlying the increase in fusion remains to be determined but may reflect a suppression of PI3K and/or Erk signaling following NGF-withdraw.”

ii) There are multiple studies showing that neurotrophin signaling modulates mitochondrial functions apart from mitochondrial dynamics and Ca^2+^ handling, including ATP synthesis, membrane potential, respiratory capacity and others, some of which are cited by the authors themselves. Moreover, mitochondrial dynamics is intimately linked with mitochondrial functions. Given that it seems likely that some of these functional parameters could also be changed, so in addition to their smaller size the mitochondria in the new NT driven steady state could also be functionally distinct and more active. At minimum, the membrane potential can be relatively straightforwardly assessed using TMRM staining, but a more thorough investigation of other parameters like ATP production and respiratory capacity will be very informative.

Prior work from the Hollenbeck laboratory using the same neuronal chicken sensory population as in the current study has reported a slight hyperpolarization of axonal mitochondria in response to NGF treatment through measurements of TMRM intensity (Verburg and Hollenbeck, 2008; referenced in original submission). We have analyzed the ATP/ADP ratio in axons using the ratiometric sensor Perceval-HR and find no effect of NGF treatment during the first 15 min (we have not addressed later time points), although both inhibition of oxidative phosphorylation or glycolysis decreases the ATP/ADP ratio as positive controls. We are not including these data in the current manuscript and hold that an analysis of the effects of NGF on axonal bioenergetics is an additional issue, which we are however considering as our work progresses.

iii) The authors have thoroughly analyzed the effects of exogenous NGF applications on mitochondrial fission, but it is important to assess what is the contribution of endogenous NT signaling. The authors can use function blocking antibodies and or pharmacological blockade of NT-Trk signaling to ask if the baseline level of fission/fusion balance is influenced. Given the finding of Blaniped's group that CamKII signaling is dynamically involved in the baseline maintenance of fission/fusion balance in addition to its role in LTP induction in hippocampal neurons, it will be interesting to asses CaMKII's role in the context of the current study.

For consideration of the issue of CaMKII we refer the reviewer to the prior response regarding the work of Divakaruni et al., 2018. Our system involves sensory axons not contacting other axons and not in an integrated network (as sensory neurons so not synapse on one another), on laminin coated substrata that specifically allows TrkA positive axons, and no other sensory subpopulations, to extend (as initially presented in Results). To our knowledge there is no “endogenous NGF signaling”, although laminin may well signal to mitochondria. The latter, while of interest, we consider to be beyond the scope of the current work.

iv) Since neurotrophins signal through both Trk and P75NTR receptors it is standard practice to assess what is the contribution of each class of receptor to the observed phenomenon. The authors do show that classic Trk mediated pathways are involved, but this does not automatically rule out p75NTR involvement, which can also augment Trk signaling in some contexts or antagonize it in others. This will be worthwhile to be assessed in the maintenance phase as well. As above, function blocking antibodies against Trk and p75NTR and pharmacological means (K252A) can be used to address this question.

The initial data indicating the effect of NGF is mediated by TrkA were the NGF concentration profile response and the well-established activation of TrkA of PI3K and Mek-Erk signaling. In agreement with the reviewer’s suggestion we have further probed the role of the TrkA receptor using two approaches. First, we used k252a and find that it inhibits NGF-induced decreases in mitochondria length. Second, we treated the cultures with 100 ng/mL BDNF, which binds the p75 receptor but not the TrkA receptor, and find that it does not induce changes in mitochondria lengths. These data are now presented in Figure 1—figure supplement 2K, and the associated text in the Results is as follows; “To further address the issue we pre-treated with the TrkA inhibitor k252a prior to NGF treatment resulting in inhibition of the NGF-induced decrease in mitochondria length (Figure 1—figure supplement 2K). […] Collectively these data indicate the effects of NGF on axonal mitochondria are mediated through the TrkA receptor.”

The results of these two additional experiments are consistent with the requirement for TrkA activation by NGF and the insufficiency of p75 binding by neurotrophins in inducing mitochondria fission and discussed on 495-501; “The concentration profile of NGF induced fission on axonal mitochondria is consistent with activation of the TrkA receptor (Kaplan et al., 1991) that activates both PI3K and Erk signaling. The observations that inhibition of TrkA using k252a inhibits NGF-induced fission and that treatment with BDNF at a concentration expected to activate the p75 receptor does not induce fission indicate that TrkA is the major receptor mediating the effect of NGF on mitochondria fission. However, a role for p75 in modulating TrkA signaling cannot at present be ruled out.”

Reviewer #3:[…] 1) For all genetic and pharmacologic manipulations – are they having effect on neuronal survival and cause degeneration? The author is using high levels of drugs such as Lantrunculin that is toxic to neurons but there is no control experiment measuring the effect of those drugs on neuronal viability in the current DRG system.

The major sign of axonal toxicity is the beading and fragmentation of the axon as microtubules undergo depolymerization. We have not seen this in any of our manipulations including the prolonged treatments with LY294002 or Mek-Erk inhibitors. Toxicity and cell death usually also correlate with the induction of mitochondria fission, which the reagents used in the study instead block.

We have used Latrunculin extensively in our prior work using chicken sensory neurons and do not find it to be toxic. Furthermore, the treatment times used in the study are relatively brief. Indeed, latrunculin has been used in numerous studies using a variety of neurons and toxicity is not, to our knowledge, reported. For example, latrunculin treatment for 24 hrs promotes the formation of axon-like processes from hippocampal neurons (Bradke and Dotti, 1999, Science 283(5409):1931-4).

Similarly, expression of DNDrp1 was not found to have evident toxic effects in the context of our studies, and similar manipulations of Drp1 have not been reported to have toxic effect in neurons until extended time periods of weeks to months and on shorter terms are neuroprotective in injury/toxic scenarios. As the literature is large and overall consistent using a variety of methods to inhibit Drp1 just a couple of examples are referred to herein: Expression of DN-Drp1 for 4 days does not have reported toxicity in cultured hippocampal neurons (Li et al., 2004; Divakaruni et al., 2018).

In conclusion, based on our prior experience working with these reagents, lack of signs of toxicity (e.g., axon beading/swelling), their long history of use in the field and characterization, and the overall literature, we do not have reason to consider that toxicity is an issue, again emphasizing the treatments inhibit fission and fission is often a characteristic of cell death and toxicity.

2) Figure 1—figure supplement 2G-H are discussing the effects of a "local" perfusion of NGF. However, as those experiments were not done under any separation or isolation of specific parts of the axon. The authors should either do those experiments in a compartmental chamber or by using labeled NGF so it can be interpreted as local effect.

In addition to the originally presented data sets involving local microperfusion of distal axons through pipette-mediated delivery of NGF (Figure 1—figure supplement 2G, H) we now also present data from experiments using microfluidic chambers providing evidence that axons in the NGF containing distal compartment exhibit mitochondria of shorter lengths than those growing into a compartment not containing NGF (Figure 1—figure supplement 2I).

“As an alternative approach to locally expose distal axons to NGF we used microfluidic compartmentalized chambers and measured the length of mitochondria in axons that grew into compartments containing no NGF or NGF (the cell body compartments did not contain NGF). Axons in compartments containing NGF exhibited shorter mitochondria relative to no NGF (Figure 1—figure supplement 2I), consistent with a local action of NGF.”

Added to Materials and methods;

“Microfluidics

Microfluidic chambers were generated in house as described in Sainath et al.

(2017b; see Figure 1—figure supplement 2I for an example). […] To wash out the Mitotracker dye, the media was removed from both axonal reservoirs and dye free media was added to one side of the axonal reservoir and allowed to flow over the axons to the opposite axonal reservoir.”

3) The IF imaging of Phos-DRP, Phos-ERK and Phos-AKT (Figure 4) are not in high quality, and they are the only direct evidence (not pathway inhibition) used to describe the time course of NGF effect on their neuronal system. The authors should provide experiments with higher resolution and imaging quality. Also, the authors are claiming that P-AKT and P-ERK are higher in mitochondria positive axonal segments. However, the signal enrichment in those parts after 6 minutes is only around 150%, but when the authors measured the whole axon they found an enrichment of 450%, thus contradicting their own statement.

The pErk data are from 7.5 min and 6 min after NGF treatment in Figure 4—figure supplement 1D and G, respectively. Direct comparison between the two is thus not warranted.

The reviewer is correct about the discrepancy in the pAkt data presented in Figure 4—figure supplement 1F and I. The data included herein are from an ongoing larger project addressing the issues of the time course of activation of pathways by NGF and their relationship to mitochondria. In considering the reviewer’s comment we noticed that data from Akt phosphorylation at serine 476 was used in making panel I, while it ought to have been the data from p308. We are grateful for bringing this to our attention. Figure 4—figure supplement 1I has been revised to show data from the phosphorylation of Akt at p308 as originally intended. Finally, we confirmed that Figure 4—figure supplement 1J is indeed from the p308 data set.

The images in the data sets are from experiments wherein the acquisition parameters are set to ensure that the signal remains well within the dynamic range of pixels for subsequent quantification, as described in the Materials and methods (Quantification of immunocytochemical labeling). While this results in visually suboptimal images it generates data sets amenable to digital quantification of fluorescence intensities within the dynamic range. The signal presents as puncta, and the treatments increase the density of puncta but have minor effects on their individual intensity (these data are not presented herein as they are reserved for the broader study of signaling along axons referred to above). The quantification shows that using these image data sets we observe the expected differences in the time course of activation of both the Akt and Erk pathways, as these have been extensively detailed through western analysis in the literature.

“NGF elevated the activity of both pathways in axon segments containing mitochondria at 6 min of treatment when fission is occurring (Figure 4—figure supplement 1G-J)”. The statement in the manuscript that the pathways are activated in axonal segments containing mitochondria, as reflected by the analysis of staining intensity at mitochondria under no NGF and NGF treatment conditions (Figure 4—figure supplement 1G and I), is not impacted by the revision.

4) The in-vivo data is referring only to the effect of DN-DRP1 on mitochondria length and density, and number of axonal branches. This experiment is discussing a long-term effect of DRP1 inhibition, with no reference to the acute effect of NGF on this system. The experiments of the acute ex-vivo explants should be also tested after administration of NGF to see that NGF administration will not cause a rescue effect, meaning the observed changes are a result of the chronicle toxicity of DN-DRP1 on the embryo.

The reviewer is correct that the in vivo experiments expressing DN-Drp1 do not address the effects of NGF but rather the normally occurring developmental branching of sensory axons in the spinal cord, a point made through the manuscript (e.g., subsection “Inhibition of Drp1 function impairs the developmental collateral branching of sensory axons in the spinal cord”; Discussion, fourth paragraph) and we did not suggest otherwise. The value of the data from the in vivo experiments, we argue and as initially stated, is to provide evidence that in a context different than NGF-mediated axon branching suppression of Drp1 function also results in the impairment of axon branching. This result is consistent with a recent study addressing the role of fission in cortical axon development (Lewis et al., 2018).

We are beginning to address the issue of in vivo induction of sensory axon branching by NGF in the context of spinal cord injury scenarios wherein endogenously generated

NGF induces the branching of sensory axons contributing to autonomic dysreflexia (collaboration with our colleague Dr. G. Smith) but these experiments are not considered for this paper and we are retaining them as a separate project. We are also currently do not have a system set up to perform the suggested experiment in the chicken embryonic spinal cord which would require work well beyond the time provided for revisions as the system would need development (e.g., best delivery method, dosage, etc) and characterization (e.g., time course of effects of exogenous NGF on axon, etc) first, a large undertaking in and of its own.

*The observed changes are a result of the chronicle toxicity of DN-DRP1 on theembryo*.

The electroporation method used results in DN-Drp1 being expressed only by a small subset of dorsal root ganglion (DRG) neurons in 2-3 DRGs unilaterally. Thus, embryonic toxicity is not a concern and we did not observe different rates of lethality of embryos after electroporation based on the construct being delivered. The axons of the transfected neurons do not exhibit signs of degeneration at the time points studied (i.e., no beading nor swelling) and although not quantified we did not observe difference in the number of transfected axons in the spinal cords between embryos transfected with control or DN-Drp1, indicating no deleterious effects. Furthermore, as noted in a prior comment, expression of DN-Drp1 or otherwise prolonged suppression of mitochondrial fission in neurons in vivo or in vitro does not have toxic effects until very delayed time periods. For example, in the most directly related other study to the current study (Lewis et al., 2018) cortical axons with In Utero Electroporation shRNA-mediated depletion of MFF from E15.5-P21 days developed normal axons that were however deficient in branching.

5) Figure 7 – The local synthesis results are very promising but require many crucial controls.a) What is the effect of protein synthesis inhibition (puromycin, anisomycin, cycloheximide) on mitochondria fission and axonal branching of the sensory neurons?

NGF-induced branching: We previously published that inhibition of intra-axonal protein synthesis inhibits the NGF-induced branching based on experiments such as those suggested by the reviewer (Spillane et al., 2012), providing the background and rationale for the current experiment on cortactin as we have shown this protein undergoes axonal translation in response to NGF (Spillane et al., 2012; Ketschek et al., 2016). We have modified the manuscript to make these previously established points clearer (added text is shown in red):

Abstract: “Fission is also required for NGF-induced mitochondria-dependent intra-axonal translation of the actin regulatory protein cortactin, a previously determined component of NGF-induced branching.”

Introduction – “During NGF-induced branching mitochondrial respiration serves to locally drive axonal actin dynamics and intra-axonal translation of actin regulatory proteins required for NGF-induced branching (Ketschek and Gallo, 2010; Spillane et al., 2012, 2013).”

First sentence of the section titled “Inhibition of Drp1-mediated fission impairs the NGF-induced intra-axonal translation of cortactin”)“The induction of axon branches and increase in the formation of axonal actin patches by NGF are dependent on NGF-induced intra-axonal protein synthesis of Arp2/3 regulators and Arp2/3 subunits (Spillane et al., 2012, 2013).”

NGF-induced fission: We have not previously addressed whether inhibition of protein synthesis might impact the induction of fission by NGF. Therefore, using the same experimental paradigm used in prior studies to address the role of axonal protein synthesis in NGF-induced axon branching (Spillane et al., 2012), we analyzed the effects of pre-treatment with the translational inhibitor cycloheximide on NGF-induced mitochondria fission and the data are presented in Figure 7—figure supplement 1B.

Cycloheximide did not have any effect on NGF-induced fission.

“We addressed whether protein synthesis is required for NGF-induced mitochondria fission. Pre-treatment with the translational inhibitor cycloheximide, which blocks NGF-induced axonal protein synthesis and branching (Spillane et al., 2012), did not impact the change in mitochondria length induced by NGF (Figure 7—figure supplement 1B). Collectively the data indicate that while NGF-induced mitochondria fission contributes to establishing NGF-induced intra-axonal translation the latter is not required for the former.”

Also, a protein synthesis inhibition should be used as a control to see the baseline signal of no-recovery after photo-bleaching.

We have previously published FRAP experiments in the context of NGF-induced cortactin translation using the myrGFP reporter system used in this work and performed the control of treatment with translational inhibitors suggested by the reviewer (Spillane et al., 2012), as is standard for such experiments. However, we agree that the same control would benefit the current manuscript by providing a direct visual and quantitative comparison to the effects of inhibiting fission of cortactin translation using the reporter system and additional data are now included in Figure 7B along with the relevant analysis, the latter in the figure legend (below):

“The NGF+CHX (35 µM cycloheximide 30 min pre-treatment before NGF treatment as in Spillane et al. 2012) shows the extent of recovery due to translation independent diffusion or transport of the myrGFP reporter from the axon proximal to the region of bleaching. The recovery in the NGF+CHX and NGF+mDivi-1 groups was not different at any time point using Bonferroni post hoc time-matched multiple comparison tests.”

b) What is the effect of AKT or ERK pathway inhibitors on the local translation of cortactin?c) Is there endogenous translation of cortactin after NGF treatment?

B and C and considered collectively as they are linked comments. We have previously published that the local translation of cortactin in response to NGF requires PI3K-AktmTOR signaling and that this pathway is required for NGF-induced axon branching (Spillane et al., 2012).

We have previously shown that the increase in axonal cortactin *protein* levels in response to NGF is dependent on axonal translation (Spillane et al., 2012; Ketschek et al., 2016). Following the reviewer’s suggestion, we now present additional data detailing the increase in cortactin protein levels in axons in response to NGF (we note that this was also originally shown in Figure 7C, D) and that the increase is attenuated by inhibition of Mek-Erk signaling (Figure 7—figure supplement 1A).

“We have previously reported the requirement for PI3K signaling in the

NGF-induced intra-axonal translation dependent increase in axonal levels of cortactin (Spillane et al., 2012). Inhibition of Mek-Erk signaling attenuated the NGF-induced increase in cortactin axonal protein levels (Figure 7—figure supplement 1A).”